# Sequential stacking link prediction algorithms for temporal networks

Xie He [1], Amir Ghasemian [2], Eun Lee [3], Aaron Clauset [4,5,6] & Peter J. Mucha [1]✉

Link prediction algorithms are indispensable tools in many scientific applications by speeding up network data collection and imputing missing connections. However, in many systems, links change over time and it remains unclear how to optimally exploit such temporal information for link predictions in such networks. Here, we show that many temporal topological features, in addition to having high computational cost, are less accurate in temporal link prediction than sequentially stacked static network features. This sequential stacking link prediction method uses 41 static network features that avoid detailed feature engineering choices and is capable of learning a highly accurate predictive distribution of future connections from historical data. We demonstrate that this algorithm works well for both partially observed and completely unobserved target layers, and on two temporal stochastic block models achieves near-oracle-level performance when combined with other single predictor methods as an ensemble learning method. Finally, we empirically illustrate that stacking multiple predictive methods together further improves performance on 19 real-world temporal networks from different domains.

Real-world network data are often incomplete for a variety of reasons, such as missing relations in social networks[1]; inherent noise and expensive, tedious, and time-consuming data collections in biological networks[2]; and specific user privacy limitations in wireless networks[3]. Link prediction algorithms have been proposed in many different disciplinary settings, including social[4], biological[5], information[6], and epidemic[7] networks as a way to either impute missing connections or guide the use of limited resources for link measurement. Among methods for link prediction, features that statistically summarize network structures (hereafter referred to as "topological features") have been well studied and are widely used for prediction on static networks[8]. Popular topological features include common neighbors[9] and the clustering coefficient[10] for social networks, Katz centrality, resource allocation for electrical power grids and protein-protein interaction networks[11], and the local path index[12] for airport transportation networks[11]. In static networks, recent work[13] has shown that combining[14] multiple topological features using a meta-learning algorithm such as stacked generalization can produce near-optimal link prediction results.

Whereas the recording of many real-world networks has been previously limited to a single snapshot in time, modern methods of data collection and newer sources of network data (e.g., social media and other digitally tracked interactions) increasingly provide detailed temporal information on how connections change over time[15]. The temporal changes in these network datasets include measurable variations in their topological features over time[16,17], and exploiting these time-varying correlations should make link prediction algorithms more accurate in this setting[18–20]. Here, we focus on the temporal link prediction task of identifying missing connections in a network at a given point in time by leveraging data from earlier times. Accurate temporal link prediction has many real-world applications[21], including recommendation systems on social media[4] and the prediction of brain

[1]Department of Mathematics, Dartmouth College, Hanover, NH, USA. [2]Yale Institute for Network Science, Yale University, New Haven, CT, USA. [3]Department of Scientific Computing, Pukyong National University, Busan, South Korea. [4]Department of Computer Science, University of Colorado, Boulder, CO, USA. [5]BioFrontiers Institute, University of Colorado, Boulder, CO, USA. [6]Santa Fe Institute, Santa Fe, NM, USA. ✉e-mail: peter.j.mucha@dartmouth.edu

activity[22]. Like static link prediction, approaches have been developed for temporal link prediction problems[21], including probabilistic modeling[23,24], matrix and tensor factorization[18,25], spectral clustering[26], network embedding[27–29], and deep learning[30].

Generally speaking, temporal link prediction methods can be classified into four main categories: matrix factorization and probabilistic-based techniques[18,23–26], time series-based techniques[31], embedding and neural network-based algorithms[27–30], and extending static topological measures to temporally-varying networks for prediction (hereafter referred as "temporal topological features"). Each category has its own strengths and weaknesses. Tensor factorization can capture both local and global features and transitional patterns in dynamic networks, but it is not scalable for large graphs due to its high computational complexity. Time series-based techniques can effectively capture the dynamics of networks but struggle with capturing non-linear temporal patterns. Deep-learning-based algorithms can capture transitional patterns, but they may lack interpretability for feature selection. Temporal topological features can be both effective and interpretable, but their definitions are dataset-specific and their calculations are computationally expensive[21].

Among these, temporal topological-based methods are particularly applicable for a wide variety of real-world applications because of their interpretability. This is exceptionally important for social and biological network analysis to understand the reasoning behind the results[21,32,33], in particular as domain experts may find it hard to understand and trust complex models like neural networks because of the lack of intuition and explanation in their predictions[34,35]. In contrast, using topological properties of networks as features in the training and testing largely increase the interpretability of the model,

and will help explain how the results from the prediction were generated. For example, Noulas et al.[32] study a human mobility network and propose a link prediction approach influenced by temporal variation and other features of the network. Ibrahim and Chen[33] use eigenvector centrality while incorporating temporal dependence to help with link prediction in social networks. However, because of their ambiguous definition and high computational cost, they require considerable additional feature engineering work[16] compared to their corresponding static features[36] (see also Results).

To address these limitations, we want to (i) find replacement features for temporal topological features in temporal link prediction that are fast, accurate, scalable, and retain the interpretability of the temporal topological features; (ii) while retaining the interpretability, design an ensemble learning framework that could learn to improve itself based on different predictors.

To solve the first problem, we propose an approach that replaces temporal topological features with static topological features in temporal link prediction. We extend the static network stacking method proposed by Ghasemian et al.[13] to temporal networks by stacking static features from multiple sequential temporal layers (hereafter referred to as sequentially stacked features). We construct a sequential stacking link prediction framework (diagrammed in Fig. 1) using a multilayer representation for temporal networks[37], i.e. each layer describes interactions at a corresponding time or time period (a window), and utilize 41 static topological features. In this algorithm, we use two important parameters: the "search variable" $u$ (which we default to $u = 6$ throughout the paper), giving the total number of layers back in time that we will consider for training the predictor, and the "flow variable" $q$ (which we default to $q = 3$ throughout the paper) to be the

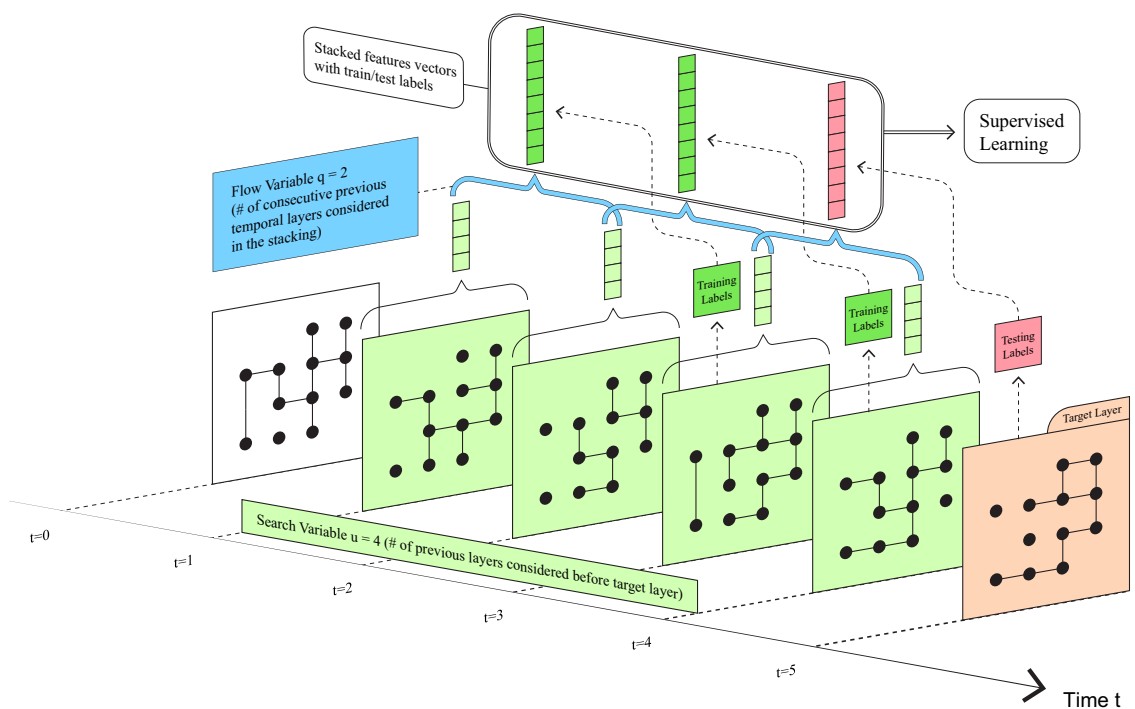

**Fig. 1 | A diagrammatic explanation of the sequential stacking approach for link prediction in temporal networks.** We use $q$ consecutive temporal layers in a stacked feature vector to predict links in the target layer; we call $q$ the "flow variable" (blue: $q = 2$ in the diagram; $q = 3$ throughout all of our experiments here). We train the prediction (see Methods) using $u$ layers before the target ($u > q$); we call $u$ the "search variable" (green: $u = 4$ in the diagram; $u = 6$ throughout all of our experiments here). Features are generated for sampled dyads (node pairs) in each layer and stacked across $q$ consecutive layers, with edge presence/absence labels from the following layer (green for training and red for testing). We then use

standard supervised learning algorithms to train and generate link predictions in the target layer. As diagrammed here, no network information is used from the target layer, only the edge presence/absence labels (the "completely-unobserved setting"). When we consider the "partially-observed setting", the sequentially stacked features include static topological features as calculated from the partially-observed target layer (throughout our partially-observed experiments here we 5-fold cross-validate the target layer, uniformly sampling 80% of node pairs to predict links on the remaining 20%).

number of consecutive temporal layers considered together in the stacking of features across time (see Fig. 1 and Methods for detailed descriptions). In place of computationally expensive temporally-extended definitions of static features, our sequential stacking approach computes less expensive static features within each layer individually from sampled dyads (see "Sampling dyads for training and testing" in Methods), and then stacks them together to form feature vectors that capture the temporal variation of the network. Since not all of the temporal topological features are well-defined, we compare the sequentially stacked static features with 5 defined temporal topological features; to further expand our comparison set, we further compute ARIMA-based time-series predicted topological features for all of the 41 static features and compare their performance with that from the sequentially stacked features. We find in both comparisons that sequentially stacked features are at least as accurate, or better, in temporal link prediction tasks while also being both computationally more efficient and conceptually better defined.

To solve the second problem, we extend the framework for the sequentially stacked features to construct a meta-learning framework for ensemble sequential stacking. The benefits of this method include that the sequentially stacked topological features are easy to calculate and interpret, while also incorporating possible accuracy enhancements from other methods (e.g., neural network, matrix factorization, time series methods, etc.). For clarification, hereafter we refer to sequential stacking with topological features alone as Top-Sequential-Stacking, and stacking with additional predictors as Ensemble-Sequential-Stacking.

The task of temporal link prediction encompasses prediction scenarios that are more general than those of link prediction on static networks. For instance, the standard setting for static link prediction requires partial observation of the static network, while in temporal link prediction the target layer within which we make predictions of missing links, e.g., a future period of time, might be missing entirely[21], and the task is to predict that entire layer using information from the previous layers[4,38], or the target layer might be partially observed and the task is to predict the missing portions of that layer. We compare and contrast our framework under the two different settings of temporal link prediction: (i) a completely-unobserved target layer and (ii) a partially-observed target layer (see Methods). Whereas some methods for temporal link prediction are not applicable when the target layer is completely unobserved[39,40], both Top-Sequential-Stacking and Ensemble-Sequential-Stacking perform well in both partially observed and completely unobserved settings.

We benchmark our Top-Sequential-Stacking and Ensemble-Sequential-Stacking approach against a tensorial method based on stochastic block modeling of multilayer networks[39] (denoted as "Tensorial-SBM"), a network embedding deep learning framework, E-LSTM-D[29], and an adapted supervised time-series ARIMA model prediction[40] (denoted as "Time-Series") (see Methods for detailed implementation and parameter choices).

We test performance across a rich space of synthetic networks constructed from two variants of degree-corrected multilayer temporal stochastic block models (T-SBM): a multilayer SBM with community label change over time ("community-label T-SBM") and an "edge-correlated T-SBM"[41,42] (see Methods, and see Supplementary Information (SI)). For each of these probabilistic synthetic network models, we carry out an analytical calculation similar to Ghasemian et al.[13] to estimate the maximum average predictability achievable by an oracle that knows the underlying temporal random graph model (i.e., the ground truth probabilities that govern whether each pair of nodes is connected), and we demonstrate that Top-Sequential-Stacking and Ensemble-Sequential-Stacking method achieve oracle results on these synthetic networks compared to this oracle-level performance (see Results and SI). We then evaluate its performance on 19 real-world temporal networks from social, technological,

transportation, and biological domains. We observe that, between them, Top-Sequential-Stacking and Ensemble-Sequential-Stacking achieve the highest accuracy on 16 of 19 real-world datasets in the completely-unobserved setting and on 17 of 19 in the partially-observed setting and, in the remaining cases, we observe only minimal reduction in performance relative to the best predictor.

In summary, we find that (i) the sequential stacking of static features is a highly effective replacement for temporal topological features both in terms of performance and computational cost; (ii) the Top-Sequential-Stacking and Ensemble-Sequential-Stacking learning approaches to temporal link prediction are highly accurate with low computation cost and good theoretical interpretability, with Ensemble-Sequential-Stacking able to incorporate other predictors for higher performance. Furthermore, our T-SBM synthetic network experiments achieve solid results compared to those achievable by an oracle that knows the underlying model that generated the probabilistic synthetic networks. Both Top-Sequential-Stacking and Ensemble-Sequential-Stacking demonstrate strong performance across various types of real-world and synthetic networks. By providing these methods as open-source code, scientists in various application domains can access an accurate, efficient, and interpretable tool for temporal link prediction. We conclude with a discussion of limitations and possible future directions.

## Results

We first consider a set of temporal topological features utilized in previous studies on link prediction. By contrasting results between 5 different sequentially stacked features and temporal topological features, we can assess whether including the temporal features improves link prediction accuracy on temporal networks and, if so, assess the induced trade-off between accuracy and computation time. We then further consider the entire set of 41 topological features and compare the results between sequentially stacked features and temporal features constructed by the time-series ARIMA model to show that sequential features outperform in both computation time and performance. We demonstrate that the sequential stacking approach achieves near-oracle-level performance under varied synthetic network models and prediction settings, and strong performance on real-world network data.

### Temporally-extended versus sequentially-stacked features

The definitions of temporal topological features can require considerable additional feature engineering work[16] compared to their corresponding static features[36] in part because a single static topological feature can often be temporally extended in multiple reasonable ways[16]. For example, temporal closeness centrality is defined differently in Refs. 43 and 44. At the same time, temporal topological features typically require significant computational resources, e.g., graph path-based temporal topological measures[16] can have long computational times with high memory usage for moderately-sized temporal networks. Indeed, even for sparse graphs with $m$ proportional to $n$, the temporal betweenness centrality definition from Zaoli et al.[45] has an $O((n^2\ell)^2 log(n^2\ell))$ time complexity, which would be prohibitively expensive for larger networks with many temporal layers (here $m$ stands for the number of edges, $n$ stands for the number of nodes, and $\ell$ stands for the number of layers in the temporal network).

On the other hand, ignoring temporal variation in topological features could potentially lead to inaccurate or even biased predictions[46]. Temporal topological features have been useful in the analysis of temporal networks in several different settings. For example, Sett et al.[47] show that using temporal features resulted in higher accuracy scores for temporal link prediction in multi-relational networks; Muniz et al.[48] demonstrate that unsupervised link prediction on social networks can be improved by combining topological, temporal, and contextual information; and Przytycka et al.[20] argue that the shift

from static to dynamic network analysis could be useful in identifying the temporal and contextual signals underlying cell interactions, which is essential for understanding biological cell networks because proper cellular functioning requires huge amounts of precise time and coordinate information. Temporal topological features can also be important for answering ecological and evolutionary questions about how biological networks change over time, and under-utilizing them can result in incorrect conclusions[46]. Together, past work on temporal topological link prediction algorithms suggests real benefits via improved accuracy, but at a potentially high or even prohibitive cost, and it remains unclear how to balance those factors in applications.

In contrast, a wide variety of topological features are well-defined and computationally efficient for static networks[49], and exhibit good performance in (static) network link prediction tasks[13,50]. Consequently, static network features are often used in temporal link prediction[51], even as completely ignoring the temporal nature of the data can lead to biased outcomes in temporal link prediction tasks[46].

Here, we explore the trade-off between the high computational complexity of temporal features and possible decrease in performance using only static topological features by developing and systematically evaluating a sequentially stacked link prediction algorithm for temporal networks that combines multiple static topological features within a temporally-aware meta-learning framework to produce improved temporal link predictions. This approach is motivated by the (static network) stacking approach in Ghasemian et al.[13], which we extend to temporal networks by stacking static features from multiple sequential temporal layers, which we refer to as sequentially stacked features (see Methods).

For this subsection, all experiments are conducted in the partially observed setting, with sequential stacking of specially selected and identified features for comparison with using the analogous temporally-extended features. Performance with the full set of 41 sequentially-stacked features is discussed in depth later in the Results. We purposefully consider the stacking of the selected features here to show that sequential stacking, even of very simple static features, is both computationally cheap and at least as accurate for link prediction as using a corresponding temporally-extended feature or a time-series auto-regressive feature constructed by the ARIMA model. We demonstrate the results for temporal topological feature comparison on a subset of small synthetic networks generated specifically for saving computational time, since some of the temporally-extended features are particularly expensive to compute for larger networks. We further demonstrate the comparison with time-series ARIMA temporal features on all of our datasets to show that sequential features are indeed better both in terms of

computational cost and performance of individual features for all of the 41 features (see also SI).

**Common neighbors.** As an illustrative example, we start by evaluating a simple topological feature that has been widely utilized in temporal network analysis[52] and for link prediction in different domains[9,50]: common neighbors. We define node $j$ to be temporal common neighbors of node $i$, if node $j$ has been neighbors with node $i$ in all of the $q$ temporal layers prior to the target layer of interest. In Fig. 2, we compare link prediction performance using temporal common neighbors versus stacking the sequence of common neighbors across temporal layers. We reiterate that, in this experiment, we do not apply the sequential stacking of the full set of 41 features, but rather only the stacking of common neighbors, which allows a careful exploration of performance differences between sequentially-stacked versus temporally-extended features.

We perform these experiments on 19 real-world datasets from different domains and 90 synthetic networks from two degree-corrected temporal stochastic block models ("T-SBM"; see Methods): the "community-label T-SBM" incorporates a node-level correlation in the assigned community labels that generates temporal variation in the community structure[25,41] while the "edge-correlated T-SBM" keeps the community labels the same through time but includes an edge correlation between neighboring temporal layers[42]. We measure link prediction performance by the area under the receiver operating characteristic curve (AUC)[53] (see Methods and SI).

This experiment indicates that the sequential and temporal common neighbors achieve similar link prediction performance on many real-world datasets (panel a Fig. 2), with sequential common neighbors slightly out-performing its temporal counterpart on several others; but we also observe several cases where temporal common neighbors fail to provide any useful information for link prediction (i.e., AUCs near 0.5). These findings are recapitulated in our experiment using the T-SBM datasets, in which we find that sequential common neighbors succeed at link prediction in the community-label T-SBM cases whereas the temporal common neighbors almost always produce AUCs near 0.5 (panel b Fig. 2). Noting that this temporal common neighbors definition ignores all variation across sequential layers, the inability of this feature to provide useful information about how node assignments change over time in the community-label T-SBM is perhaps not surprising, whereas sequentially stacking static common neighbors explicitly tracks how these features changes across time.

In contrast, our experiments with edge-correlated T-SBM networks (panel c Fig. 2) show similar AUC scores for both temporal

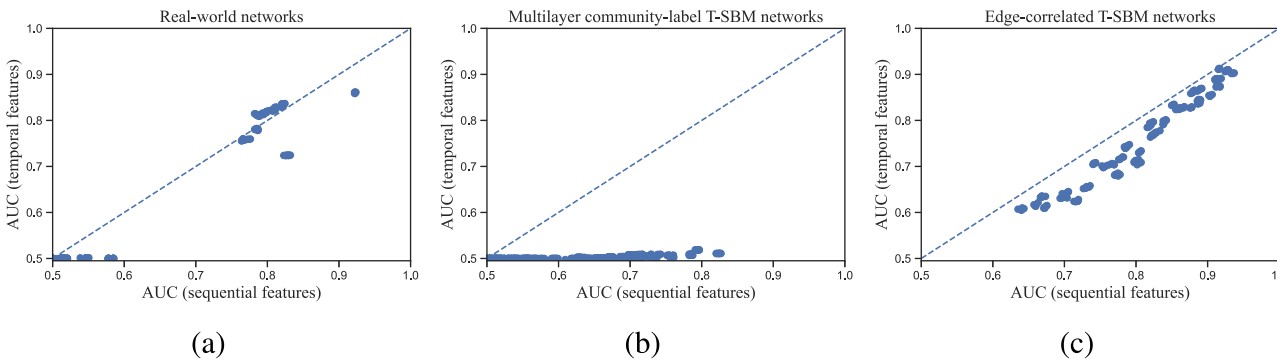

**Fig. 2 | Comparison of link prediction performance in the partially-observed setting using temporal common neighbors (vertical) versus sequential common neighbors (horizontal).** Performance is quantified here by the area under the receiver operating characteristic curve (AUC), from each run over 5-fold cross-validation on the target layer, on each of **a** 19 real-world temporal networks, **b** 45 random realizations from degree-corrected community-label temporal stochastic block models (T-SBM) at different parameters, and (**c**) 45 random edge-correlated T-SBM realizations at different parameters. (See Methods and Fig. 4 for the 45 different parameter combinations in each.) The diagonal line ($y = x$) indicates equal performance.

common neighbors and sequentially stacked common neighbors, with the latter doing better on average. Indeed, in the vast majority of cases, the use of sequential stacking of static common neighbors outperforms temporally-combined common neighbors (Fig. 2), and the few cases where temporal common neighbors give higher performance are relatively close (in contrast with the many times where using temporal common neighbors returned AUC values near 0.5).

These results illustrate the ability of the sequential stacking of static features to exploit the temporal correlations in common neighbors that aid in predicting future connections, in contrast to the simple temporal 'pooling' behavior of the temporally-extended common neighbor feature. Moreover, the sequential stacking approach is less computationally expensive than temporally-combined common neighbors and yet yields comparable or higher accuracy for link prediction in all cases.

**Experiments with other temporally-extended features.** Common neighbors is a particularly simple topological feature. Other temporal topological features are more complicated, and hence may capture more predictive information for link prediction, such as network latency and various centrality measures[43,44,54–56], the corresponding static versions of which have also been applied successfully for static link prediction[10]. Following the temporal feature definitions introduced in Thompson et al.[36] and Zao et al.[45], we consider four additional temporal features: betweenness centrality[45], closeness centrality[36], degree centrality (named temporal centrality in ref. 36), and network latency (named reachability latency in ref. 36). All of these features are computationally expensive[36,45]: even using 20 parallel processes with 8 GB memory each (on the Discovery7 cluster, see Acknowledgments), we are not able to finish the temporal features computation within a week for the larger real-world networks and our synthetic networks with 250 nodes and 10 layers. We anticipated this outcome, considering the differences between the datasets used in Thompson et al.[36] and our study. The datasets in Thompson et al. consist of 10 subgraphs that represent the dissected 264 brain regions, with approximately 26 nodes each. In contrast, our smallest networks include 50 nodes and 10 layers, with a high edge density. The computational cost of temporal features tends to grow polynomially with the numbers of nodes and edges in the network, posing challenges when applying the same method to larger networks. To obtain results despite this high computational cost, we construct 45 different synthetic edge-correlated T-SBM networks with 50 nodes and 10 layers

each (see SI Section A). Allowing the calculation to run for a week (again with 20 processes at 16 GB memory each), produces the temporal network features for 24 of these 45 smaller synthetic networks. In contrast, the calculation of all of the corresponding static features finishes within 10 minutes (0.1% of the compute time) on the same hardware (panel b Fig. 3). Despite requiring considerably more computational time across these synthetic networks, our experiments show that link prediction using these temporal extensions substantially under-performed compared to the accuracy obtained from sequentially stacking the corresponding static features (panel a Fig. 3). Notably, the temporal extensions of both closeness centrality and network latency completely failed at link prediction (panel a Fig. 3, AUCs near 0.5).

**Comparison against time series features.** In addition to temporal topological features, time-series extended features are also known for their good explainability compared to other methods[21]. To gain insights, we compared the results of these features with sequentially stacked features. Following the setup in Ozacan et al.[40], we calculated different metrics such as betweenness centrality and common neighbor scores, and applied an auto-regressive integrated moving average (ARIMA) on the previous time series data to predict future metric scores. The original paper used 12 features to form a predicted feature vector for the targeted temporal layer, and a supervised learning method such as Support Vector Machine (SVM) or Random Forest for prediction. In our study, we extended these features to all 41 features used in our method, compared single features against each other on all of the real and synthetic networks, and presented the full results in SI, Section F. However, as shown in Fig. 3, even though time-series features are computationally more efficient than temporal topological features, they still take longer to compute than sequential features. While the overall set of prediction results is similar to that for sequential features, they are not better and require a higher computational cost. The same trend is also observed in all of the experiment results presented in SI, Section F.

**Near-Oracle-level performance in two temporal link prediction settings**

The theoretically optimal performance of any algorithm can only be known when the underlying model determines the structure of the data, i.e., the ground truth is known. We investigate the aspect of the performance of the sequential stacking method by deriving the

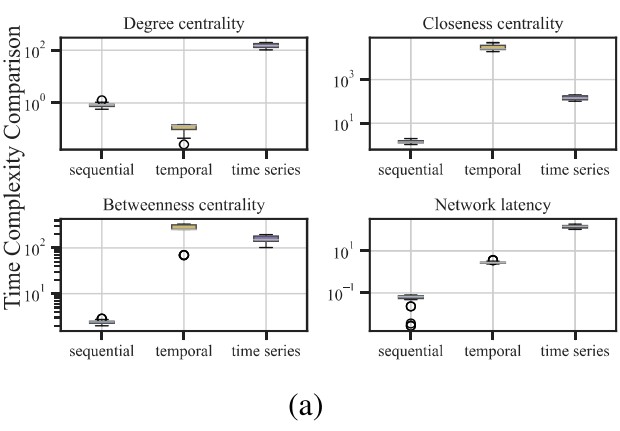

(a)

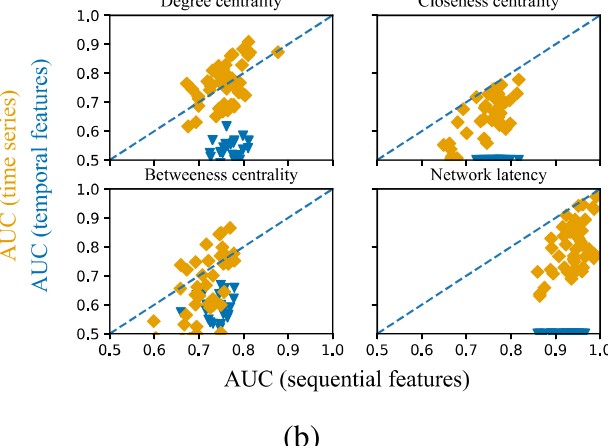

(b)

**Fig. 3 | Time complexity analysis for temporal topological features versus sequentially static features. a** Time in seconds to compute temporal-extended network features, time-series modeled temporal features, and sequentially static features for 4 different types of features on 24 random edge-correlated T-SBM realizations with 50 nodes and 10 layers each. (20 parallel processes with 8 GB

memory, see Acknowledgments) **b** Comparing AUC scores for link prediction from these temporal features (y-axis, blue) or time-series modeled features (y-axis, orange) versus the corresponding sequentially stacked features (x-axis). Both panels display data obtained in the partially-observed setting.

theoretically optimal link prediction AUC score (see SI Section D) for the community-label T-SBM and edge-correlated T-SBM (see Methods) that would be achieved by an "oracle" who knows and uses the underlying model specification to make predictions. That is, the oracle knows the particular probability $\Pr(i \to j|\theta)$, where $\theta$ is the model specification, for every pair $I, j$ in the network, which governs the ground-truth likelihood that a pair may or may not be connected in the link prediction test set.

Unlike static link prediction, which assumes a network is partially observed, with some edges missing while others are observed (i.e., for a single layer), temporal link prediction algorithms face different requirements for different applications. We demonstrate that the sequential stacking method is applicable to both partially observed and completely unobserved target layer settings (see Methods), benchmarking its performance against three state-of-the-art methods: the Tensorial-SBM method[39] from probabilistic inference on stochastic block models, the E-LSTM-D method[29] from network embedding, and the Time-Seires method from auto-correlation ARIMA on static network features[40]. In doing so, we note that some methods cannot be applied to a completely missing target layer; in particular, we benchmark with all of Time-Series, E-LSTM-D, and Tensorial-SBM for the partially observed setting, but only with E-LSTM-D and Time-Series for the completely unobserved setting, because Tensorial-SBM does not support the completely missing target layer scenario[39]. We further note that stacking is a meta-learning ensemble method that can naturally add other individual predictors to improve its own performance. As such, we also obtain results by including Tensorial-SBM (where possible), E-LSTM-D, and Time-Series predictions as additional features within the sequential stacking method.

Both temporal stochastic block models (T-SBM) considered here use the same degree-corrected SBM procedure to generate the first temporal layer, which then serves as a foundation for later layers[25] generated either with node community label correlations in the community-label T-SBM[41] or on common node community labels with edge correlations in the edge-correlated T-SBM[42]. For these experiments, we construct 45 networks with 200 nodes and 10 layers from each model (90 temporal networks in total), adjusting the parameters systematically to provide a variety of different underlying structures. Specifically, we vary the dependency between layers $p$ (community label copy rate in the community-label T-SBM, edge copy rate in the edge-correlation T-SBM), the fraction $\mu$ of uniformly-random edges in the network, and the number of communities $k$ (details in Methods).

AUC scores for link prediction on both T-SBMs are shown for the partially-observed setting in Fig. 4 and for the completely-unobserved setting in Fig. 5. Note the performance for both the Top-Sequential Stacking and Ensemble-Sequential-Stacking methods are typically close to the oracle's theoretically optimal upper bound for all edge-correlated T-SBMs and for the partially observed case for community-label T-SBMs. Even though neither Top-Sequential-Stacking nor Ensemble-Sequential-Stacking has direct information about the target layer network features in the completely-unobserved setting, they nevertheless achieve a high accuracy score for both synthetic models. The best AUCs obtained for the completely-unobserved setting are typically lower than in the partially-observed setting for the community-label T-SBM, reflecting the lack of training data on the target layer. For the edge-correlated T-SBM model, however, the performance for the completely-unobserved case remains nearly the same as in the partially-observed case. Precision-recall results for these experiments are provided in Section D of the SI.

The two T-SBMs give very different oracle upper bounds due to their different nature across the two settings (Figs. 4 and 5). In particular, the predictability is generally lower for networks with fewer communities and with higher fractions of uniformly-random edges added across the network, consistent with the findings in the single-layer cases considered by Ghasemian et al.[13]. We note that the link

predictability for the community-label T-SBM shown in both settings (panel a of Figs. 4 and 5) is on average higher than on static networks with similar community structures[13]. This behavior is consistent with the node-label temporal dependency usefully accumulating information across the previous temporal layers[57]. On the edge-correlated T-SBM (panel b of Figs. 4 and 5), the predictability is even higher than on the community-label T-SBM. This increased performance is plausibly due to the enhanced impact of the temporal dependency parameter $p$ as an edge copy probability in this model (e.g., most of the edges are simply preserved when $p = 0.8$ for the edge-correlated T-SBM).

In our analysis of both types of T-SBMs, we have found that calculating the temporal scalar auto-correlation score in Lacasa et al.[58] (see SI Section E) is an effective way to explain the predictability and AUC score of the synthetic dataset. A higher auto-correlation score often means the dataset is more predictable. Edge-correlated T-SBMs generally have higher predictability than community-label T-SBMs. We have also observed that the predictability increases with higher values of $p$ and $k$, and decreases with higher values of $\mu$, which corresponds well with the randomness of the inference model setup.

Across these experiments, we find that the Ensemble-Sequential-Stacking method outperforms the individual benchmark predictors across almost all parameters for the T-SBMs under both partially-observed and completely-unobserved target layer settings of temporal link prediction, and compares well with the theoretical oracle-level performance (Figs. 4 and 5). In addition to that, Top-Sequential-Stacking alone also gives a decent performance across different types of T-SBMs under both settings. Outside of the completely-unobserved setting for the community-level T-SBM, in most cases, the Ensemble-Sequential-Stacking achieves an almost-oracle AUC score. Moreover, across both T-SBMs and both prediction settings, we find that adding the available benchmark predictors to Ensemble-Sequential-Stacking typically yields better performance than any of the three methods by themselves, indicating that each individual method exploits somewhat non-overlapping information between observed and unobserved links. There are some exceptions where Top-Sequential-Stacking alone is able to achieve the best score without the additional predictors, further indicating the strong performance of this sequential stacking method. In particular, for E-LSTM-D and Tensorial-SBM, the improvement from incorporating topological features is typically substantial over using either of these individual benchmark predictors alone.

## Performance on real-world network data

On synthetic networks that try to model some of the key properties of real-world temporal networks, the sequential stacking approach achieves high accuracy. However, the complex nature and variety of different types of real-world networks tend to make modeling their structure a difficult task, as discussed in, e.g.,[59,60]. Methods that perform well on synthetic networks may not achieve the same level of performance on real-world networks. Using 19 real-world temporal networks from various domains (see SI Section A), we evaluate how well the Top-Sequential-Stacking performs in realistic settings, and we both compare it against the benchmark methods (Tensorial-SBM, Time-Series, E-LSTM-D), and evaluate the benchmark predictors stacked together with the sequentially stacked features, which is the Ensemble-Sequential-Stacking method.

On these networks, the Top-Sequential-Stacking alone outperforms all other individual predictors in 16 out of 19 real-world networks in the partially observed setting, and 12 out of 19 in the completely unobserved setting (Fig. 6). While in some cases for both settings, Time-Series or E-LSTM-D or Tensorial-SBM outperforms the Top-Sequential-Stacking alone, it is clear that none of the individual algorithms alone was able to stay at the peak for different real-world networks, while Ensemble-Sequential-Stacking exhibits at or near the peak performance for each temporal network. Indeed, the Ensemble-Sequential-Stacking outperforms Tensorial-SBM, Time-Series, and

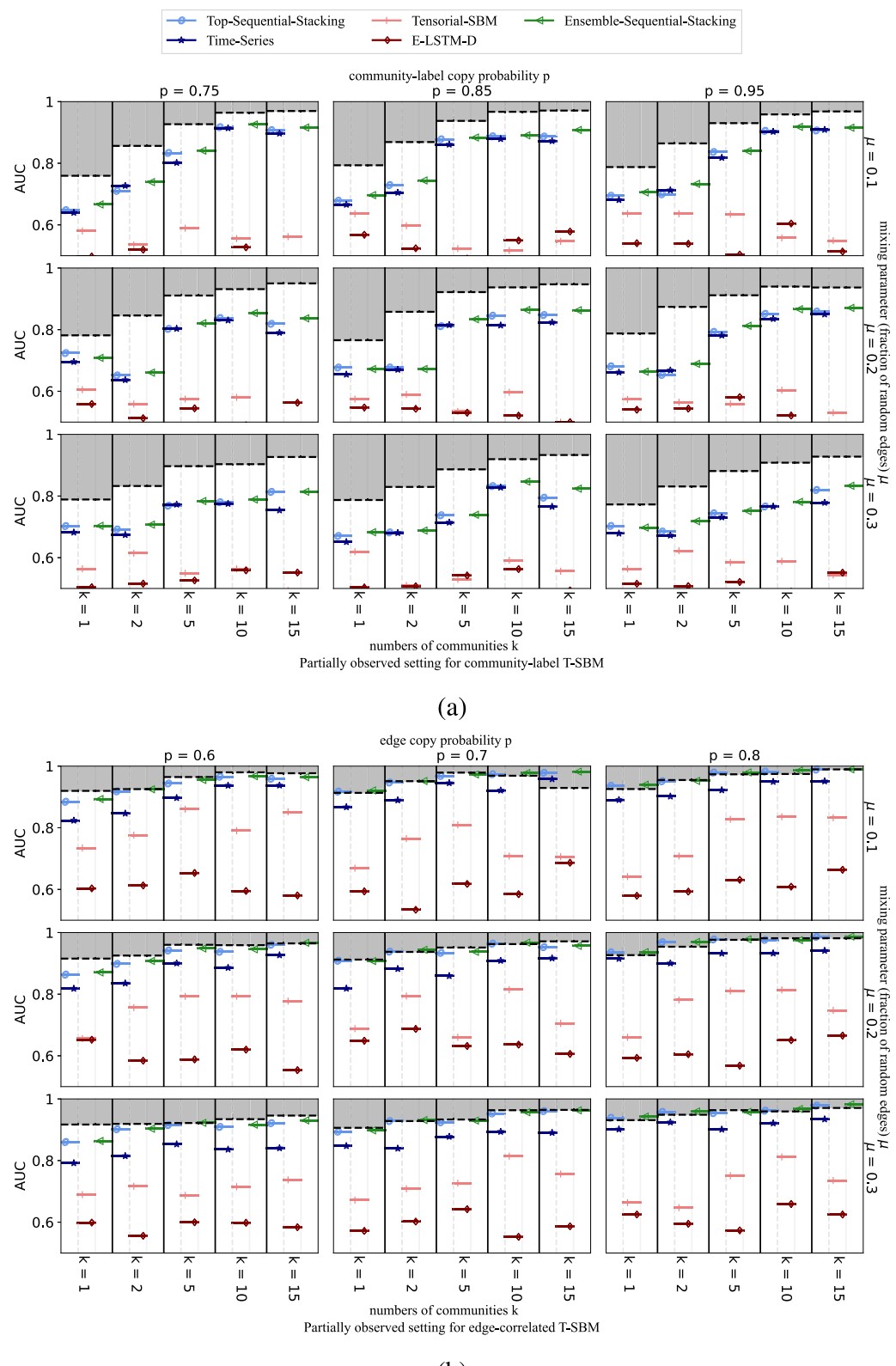

**Fig. 4 | Average AUC for partially-observed target layers on synthetic datasets.** (**a**) community-label T-SBM, and (**b**) edge-correlated T-SBM networks, each with 200 nodes and 10 layers. Each row shows results for a different fraction of random edges, $\mu$ (see Methods), at different copy probabilities, $p$ (of community labels in a, of edges in b), and numbers of communities, $k$. Dashed lines represent the theoretical maximum average link prediction performance achievable by an oracle that knows the full T-SBM specification. Colors indicate different link prediction methods (see legend). A random forest is used for supervised stacking of methods.

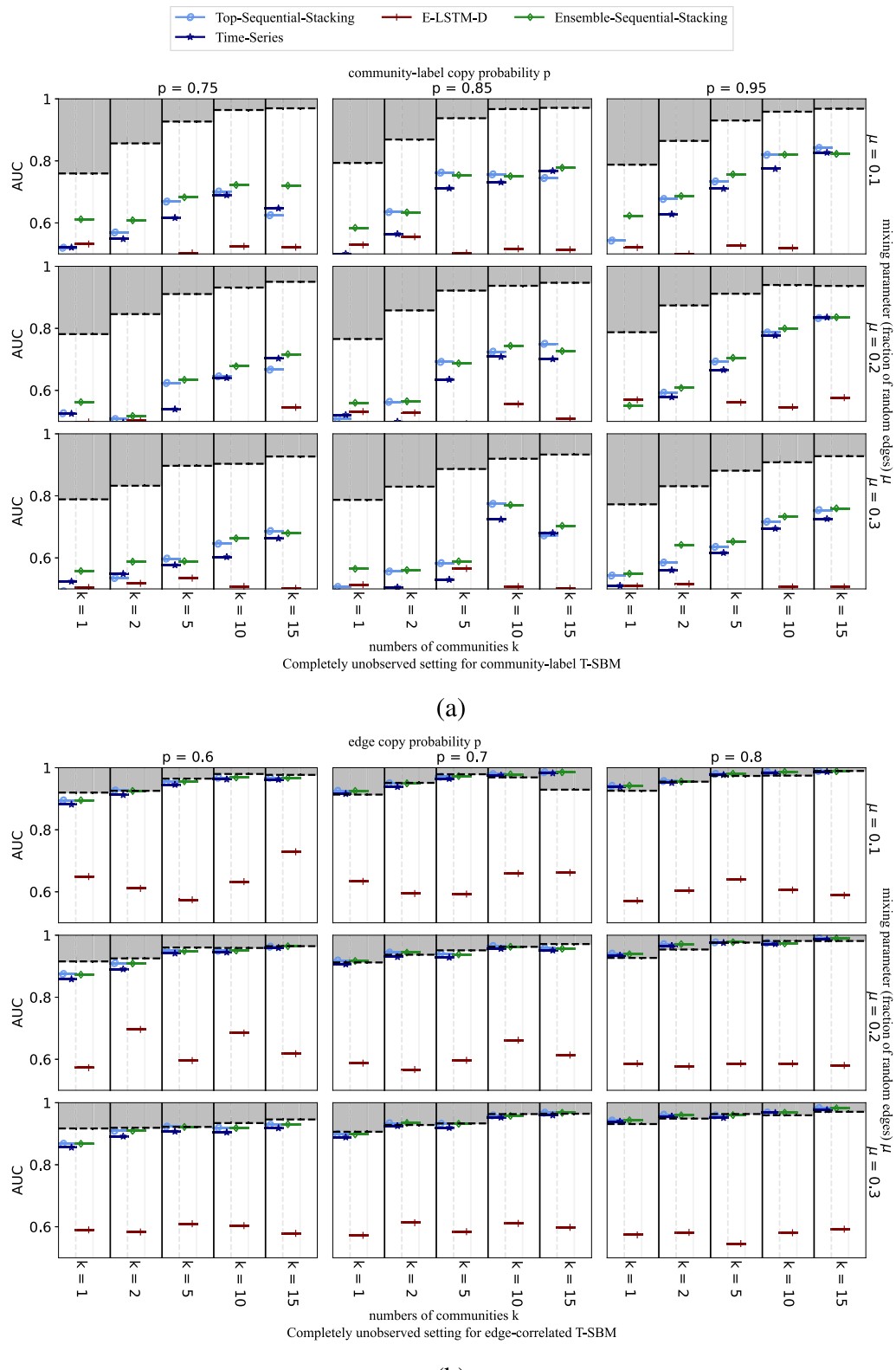

**Fig. 5 | Average AUC for completely-unobserved target layers on synthetic datasets.** (**a**) community-label T-SBM and (**b**) edge-correlated T-SBM networks, with 200 nodes and 10 layers. Each row shows results for a different fraction of random edges, *μ* (see Methods), at different copy probabilities, *p* (of community-labels in **a**, of edges in **b**), and numbers of communities, *k*. Dashed lines represent the theoretical maximum average link prediction performance achievable by an oracle that knows the full T-SBM specification. Colors indicate different link prediction methods (see legend). A random forest is used for supervised stacking of methods.

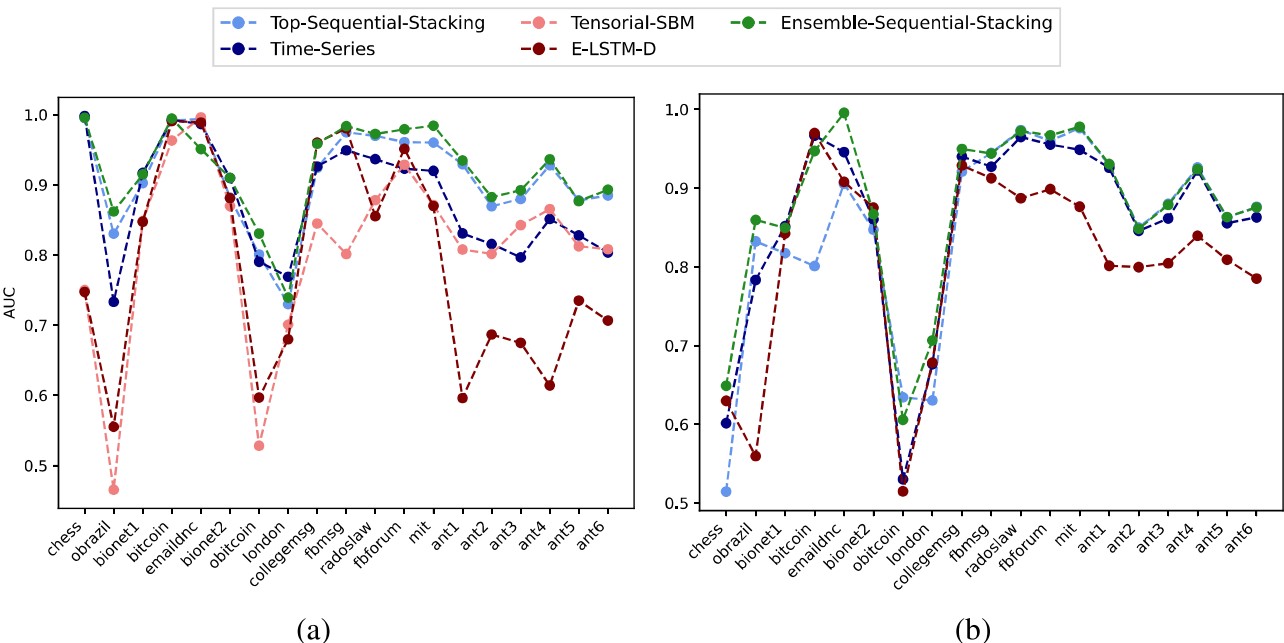

(a) (b)

**Fig. 6 | Average AUC scores for predicting links in the final layer (the target) in each of the 19 real-world networks. a** When the target layer is partially observed, we benchmark Top-Sequential-Stacking against Tensorial-SBM, Time-Series, and E-LSTM-D, and then also include these predictors with the stacking to compare against Ensemble-Sequential-Stacking. **b** When the target layer is completely unobserved, the Tensorial-SBM method cannot be used, so we benchmark against Time-Series and E-LSTM-D, and then include them as features in the stacking framework to benchmark with Ensemble-Sequential-Stacking. A random forest is used for the supervised stacking of methods.

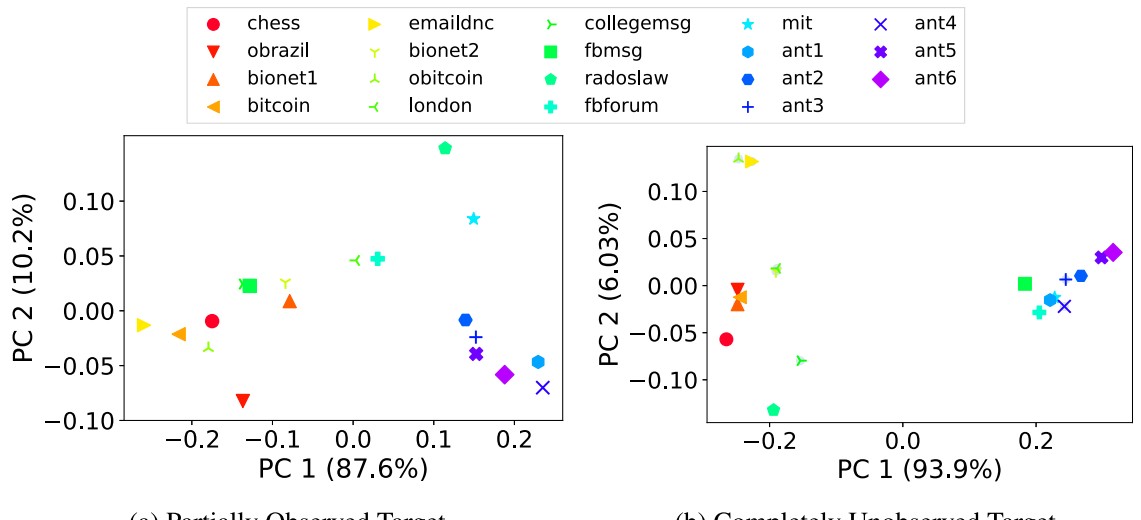

(a) Partially Observed Target (b) Completely Unobserved Target

**Fig. 7 | The first two principal components (PC) of the set of Gini importance scores of the 41 learned model features for each of the 19 real-world temporal networks, for the (a) partially-observed target layer setting and (b) completely-unobserved target layer setting.** The total accumulated explained variance ratios are marked on the figure caption respectively.

E-LSTM-D on 17 of 19 temporal networks in the partially observed case, and 10 of 19 for the completely unobserved case. Moreover, between Top-Sequential-Stacking and Ensemble-Sequential-Stacking we achieve the best performance on 16 out of 19 of the completely unobserved cases. Finally, even in the cases when they are not achieving the highest score, they are very close with only small differences in the AUC scores. In the partially observed setting, the AUC performance scores are higher, as more information from the target layer is added to the prediction problem than in the completely unobserved case. We provide precision-recall results for these experiments in Section D of the SI. The analysis of predictability can

also be linked to the calculation of temporal auto-correlation scores, as discussed in Lacasa et al.[58]. The higher the auto-correlation scores are, the higher the predictability of the real-world networks. All of the corresponding scores for the real-world networks can be found in the SI, Section E.

Recall that even a simple comparison demonstrates that link prediction with temporal and static common neighbors can perform very differently for different real-world datasets (panel a Fig. 2). We observe similarly wide variation in performance by more sophisticated methods across different real-world datasets (Fig. 6). To explore these differences, for each dataset, we computed the Gini importance score

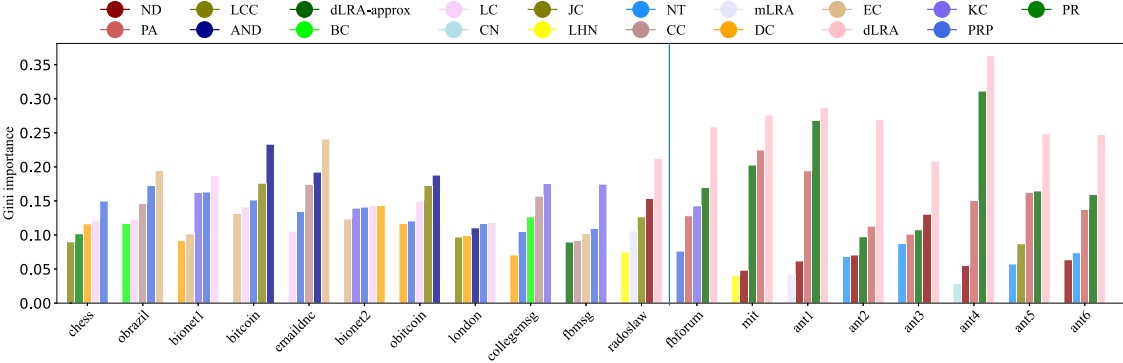

(a) Most important features (completely-unobserved setting)

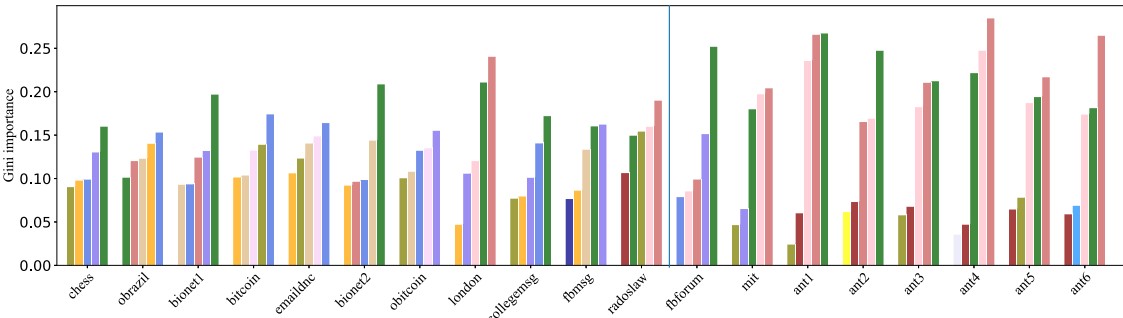

(b) Most important features (partially-observed setting)

**Fig. 8 | The 5 highest Gini importance scores for link prediction in each of the 19 real-world networks. a** shows the most important features for the completely-unobserved setting and **b** shows the partially observed setting. Acronyms for network features (detailed in SI Section A) include: PageRank (PR), the dot product of columns *i* and *j* in truncated low-rank approximation (dLRA-approx), average neighbor degree (AND), shortest path (SP), betweenness centrality (BC), degree centrality (DC), truncated low-rank approximation (LRA-approx), common neighbors (CN), low-rank approximation (LRA), Katz centrality (KC), closeness centrality (CC), the Adamic-Adar index (AA), preferential attachment (PA), eigenvector centrality (EC), resource allocation score (RA), network transitivity (NT), local clustering coefficients (LCC), Jaccard's coefficient (JC), local centralities (LC), network diameter (ND), Leichet-Holme-Newman index (LHN), average of entries of *i* and *j*'s neighbors in low rank approximation (mLRA). The vertical lines are provided to highlight the visually apparent separation in Fig. 7 of the 8 datasets on the right here from the others.

for each of the topological features used and performed principal component analysis (PCA) on these scores (separately for each setting, Fig. 7). The first two principal components for each of the two prediction settings demonstrate some distinct similarities in the high-level clustering: for example, the *ant* colony networks cluster together closely with *fbforum* and *mit* under both settings (as well as in Fig. 8, described next). This distinct group also aligns with the predictability of the temporal common neighbors (panel a Fig. 2), which further validate the diverse structures of real-world temporal networks.

The Gini importance score is a measure of feature importance used in Random Forest algorithms, which calculates the total decrease in node impurities across all the trees in the forest caused by splitting on a particular feature. A higher Gini importance score for a feature indicates that the feature has a more significant impact on the classification accuracy of the model, which makes it a useful tool for identifying the most important features in a dataset and understanding their impact[61]. Here, the 5 highest Gini importance scores for each dataset show that none of the real-world datasets share the same set of important topological features (Fig. 8). Motivated by the visually apparent clustering in Fig. 7, we use vertical lines in Fig. 8 to split the real-world networks into two groups. The group of 8 datasets to the right of the lines typically exhibits high importance for the dot product in a low-rank approximation (dLRA), its "dLRA-approx" truncation, and the preferential attachment feature (PA). In contrast, the 11 datasets on the left in the figure typically exhibit a high importance of Katz centrality (KC). Both the PCA and

the Gini importance results for temporal networks align with the NoFreeLunch theorem[62] discussed in Ghasemian et al.[13] for static networks, suggesting that the diversity of network features may mean there is also no universal best temporal link predictors, and an ensemble learning method like stacking is the only way to construct a general-purpose algorithm.

These results all demonstrate that real-world temporal networks exhibit different underlying patterns of structural organization, which leads to the drastic variation in performance of individual predictors in Fig. 6. In particular, none of the individual methods is able to achieve the best performance on all datasets.

Our results demonstrate the effectiveness and robustness of our designed approach in improving the prediction performance for temporal link prediction across various types of temporal networks. These findings validate the reliability and applicability of our method in different scenarios, providing a valuable contribution to the field of temporal network analysis.

## Discussion
Temporal link prediction and temporal network analysis, in general, have attracted greater attention recently with the increased availability of temporal network data. Indeed, most real-world networks have a temporal component, though it is not always recorded. While many measures of traditional, static network analysis have been extended to temporal networks, our results here show that these methods are not necessary for good temporal link prediction. Instead, using a

temporally aware stacking approach to combine static features generally produces superior results to using temporally-extended features as predictors. In addition to their definitional ambiguity, in which there are multiple ways to generalize a static measure to the temporal domain, temporal topological features typically also have significantly higher computational cost. This higher cost makes their mediocre performance (Figs. 2 and 3) compared to sequentially stacking static topological features particularly unappealing. At the same time, the sequential stacking of static features is typically easier to interpret[63] for trying to understand how different networks vary their structure over time (Fig. 8), making sequential stacking a powerfully useful tool for predicting missing links in temporal networks.

Moreover, our experiments suggest that our approach achieves near-oracle-level performance, especially when combined with other link prediction methods (Ensemble-Sequential-Stacking) as additional features in the stacking, on both synthetic (Figs. 4 and 5) and real-world networks (Fig. 6), and for both partially-observed and completely-unobserved target layer settings. To mimic the structural diversity of real-world networks, the two temporal stochastic block models (T-SBMs) used here are adjusted systematically to include different community sizes, community sparsity, and most importantly, temporal dependency. Our notion of optimality here derives from an oracle that has full knowledge of the random synthetic network model, and thus no algorithm can do better. In analyzing the behavior of the method on real-world networks, we find broad differences in which particular topological predictors are most boosting the overall accuracy. This extends the No Free Lunch[62] discussion of Ghasemian et al.[13] into the temporal domain, and highlights that the flexibility of the stacking approach in ensemble learning is a key feature, because it lets us understand which features are helping most for any particular network. In this way, the sequential stacking approach avoids the strong and particular assumptions that temporally-extended features make for how structure may change over time, and instead it learns those variations from the data itself, making it a more flexible model.

By developing open-source code for this stacking algorithm, we provide an efficient method that emphasizes interpretability. By offering a range of features, our method is broadly applicable and addresses the limitations of black box models commonly used in temporal link prediction. Our approach not only compensates for the computational limitations of temporal topological features but also demonstrates the potential of incorporating individual predictors to enhance model performance across structurally diverse networks in various domains.

While we have demonstrated near-oracle-level link prediction performance with sequential stacking in these synthetic and real-world datasets, applications to novel empirical domains would benefit from data-driven guidance about selecting the parameters used in the framework. Specifically, all of our results are obtained with "search variable" $u = 6$ and "flow variable" $q = 3$, that is, we use the first 6 layers to predict the 7th layer in each experiment (see Fig. 1). Since different real-world scenarios may have different temporal dependencies, how long the data from previous temporal layers may be useful for predicting future links should be expected to vary between different datasets and potentially between different temporal regimes within the same dataset. In general, larger values of $u$ and $q$ provide greater data overall for training the predictor. Meanwhile, balancing larger $u$ relative to smaller $q$ provides a larger number of cases to train on, and could be used in cross-validation for parameter tuning specified for various applications and datasets. On the other hand, increasing $u$ and $q$ may include dynamical tendencies from layers that are temporally far away from the target layer of interest, which may lower predictive accuracy. Exploring the impacts of these parameter choices in greater detail would provide better guidance for future applications and may benefit from methods for detecting change points in network data.

Temporal networks are an important and active area of research, and link prediction is just one of many temporal analysis tasks. Given its strong performance on temporal link prediction, sequentially stacked features could provide a alternative approach to other network analysis tasks on temporal networks, such as community detection, network classification, anomaly detection, and influence analysis. That is, instead of trying to further define temporal extensions of topological features for these different tasks, different approaches to sequentially stacked features might achieve better results with less computational effort. At the same time, the development of new temporal features for network analysis that are both efficient to calculate and that serve well for this and other tasks should remain an active endeavor and sequential stacking can naturally incorporate these new features in order to expand and enhance its predictive accuracy across domains and settings.

## Methods

### Temporal network notation

For concreteness, we use the notation $G = \{G_0,...,G_L\}$ to specify a temporal network with discretized time consisting of temporal layers $G_t = (V_t, E_t)$, where $t$ indexes the time points/windows with node set $V_t$ and edge set $E_t$ in that layer. In a similar way, we define the node set $V = \{V_0,...,V_L\}$ and the edge set $E = \{E_0,...,E_L\}$ to contain all layers' nodes and edges respectively. We assume that we know every node that appeared in the temporal network, i.e., we assume the full node set $V$ is given from $t = 0$ to $t = L$. The layer $G_L$ is the target layer in the temporal link prediction: we want to predict the presence and absence of edges on node pairs in $G_L$. Throughout our experiments here, we only use the last temporal layer as the target, though of course any layer might be the target provided there are sufficient numbers of layers preceding it to train a predictor.

### Parameters: search variable and flow variable

It is important for us to use as much data as possible for training the link prediction classifier. Nonetheless, we must be careful not to look back too far in order to prevent the sequential stacking feature vector from getting too long. This is because the temporal dependence of the data from many layers back could have a decreasing effect on the prediction we make in the target layer.

In balancing these needs, we set the "search variable" $u$ to be the total number of layers back in time that we will consider for training the predictor, and the "flow variable" $q$ to be the number of consecutive temporal layers considered together in the stacking of features across time (see the visualization of the framework in Fig. 1). These parameters then give us $u - q$ distinct groups of layers for training prior to the target layer.

In all of the experiments in this paper, we set $u = 6$ and $q = 3$ (and thus $u - q = 3$). That is, for each of the real-world and synthetic networks, we are using the first 6 temporal layers for training, and predicting on the 7th layer. As noted in the Discussion, future work could investigate the impacts of these choices in greater detail, especially insofar as one might expect the optimal choices to vary between datasets and possibly also within the temporal variation of a given dataset.

### Temporal link prediction: to observe or not observe part of the target layer

Link prediction on static networks usually assumes some partial observation of the network that is used to develop the predictor for the missing (unobserved) part of the network. But the general setting for temporal link prediction can be very different from that on static networks[18]. We consider two different settings for temporal network link prediction, with the network in the target layer either partially observed or completely unobserved. Sequential stacking applies to both cases in a similar manner, with the following key differences: (1) if

the stacked features include information from the partially-observed network in the target layer, this partial observation restricts which dyads are available for testing; and (2) in the partially observed setting there will be an additional group of layers available for training, formed from the labels and the additional stacking of features of the partially observed target layer, denoted here by $G_L^{observed}$.

In the completely-unobserved setting, the test labels on the dyads (node pairs) denoting edge presence/absence in $G_L$ are associated with features computed from the network layers $\{G_{L-q},...,G_{L-1}\}$. To reduce the chance of overfitting, especially insofar as edges might be expected to persist across multiple temporal layers, as well as for consistent comparisons with our partially-observed setting (described below), we split the dyads into 5 folds, with approximately equal edge counts in each fold. We then restrict our test set labels to the dyads in a selected testing fold in the target layer and restrict the training set labels to the dyads in the other 4 training folds in earlier layers. We report cross-validated results over repeated randomized folds.

For the partially-observed setting, the edge presence/absence test labels on the dyads in $G_L$ are associated with features computed from layers $\{G_{L-q},...,G_L\}$, noting that only the testing features are computed using $G_L$, while the test dyad labels are still taken from the selected test fold, thus not leaking any information about $G_L$ because the full network features are only utilized during testing. In this way, the processing of our partially-observed and completely-unobserved settings are as similar as possible, except we do not use any network information from the $t = L$ layer in the completely-unobserved setting.

### Sampling dyads for training and testing

Because of the large number of dyads in some of the real-world networks we consider, we resample to ensure testing and training on balanced classes (edge presence/absence)[64]. Specifically, we sample 10,000 edges uniformly at random with replacement to form the positive class of our dyad test set. We similarly form the negative class of our dyad test set by sampling (again, uniformly at random with replacement) an equal number of not-edges in the corresponding fold. (All networks we consider have edge density < 0.5.) Each of the edge presence/absence test labels on these dyads is associated to stacked network features (see "Dyadic features" below) computed on the previous $q$ layers (see Fig. 1), together with network features calculated from $G_L$ for the partially-observed setting (Note in particular that Fig. 1 visualizes the completely unobserved setting and so does not explicitly include these features from $G_L$.)

Our training set is similarly formed from the dyads in the 4 training folds using each of the available $u - q$ groups of layers before the target layer (plus the additional group including the training layer in the partially-observed setting), as set by the specification of search variable $u$ and flow variable $q$ (again, $u - q = 3$ in all of our results here). For each of these available groups of layers, we repeat the same procedure as for the test set, except that we sample dyads from the training folds instead of the test fold. For example, for the completely unobserved case, the furthest back in time of these training groups of layers considers sequential stacking of network measures calculated on $\{G_{L-u},...,G_{L-u+q-1}\}$, with edge presence/absence labels from $G_{L-u+q}$. In this way, the training set for the classifier is exactly $u - q$ times larger than the test set for the completely unobserved case.

For the partially observed case, the furthest back training groups of layers used for sequential stacking of network features are calculated on $\{G_{L-u},...,G_{L-u+q-1}\}$ along with $G_{L-u+q}^{observed}$, associated with edge the "observed" part of $G_{L-u+q}$ including the edges in the training folds. Note that for the partially observed case, we also sample training dyads from the observed part of the target layer $G_L^{observed}$, and thus we have an additional group of training samples with stacked features calculated on layers $\{G_{L-q},...,G_L^{observed}\}$. By this procedure, the training set is $u - q + 1$ times the size of the testing set for the partially observed case.

### Dyadic features

We extend the (single-layer) static topological stacking method of Ghasemian et al.[13] and train an edge/non-edge scoring function using network properties on each of the dyads (node pairs) on which we want to predict. That is, the topological features are calculated for each dyad, adding to the overall interpretability of our method. This approach is notably different from general neural network approaches on the entire temporal dataset, which often have less interpretable features[21]. On each layer we compute 41 static topological features, listed in the SI, combining global network measures with features calculated on each dyad and concatenations of node-level features from both participants in each dyad. That is, the resulting feature vector will have length $41q$ for the completely unobserved setting and $41(q + 1)$ for the partially observed setting.

### Random forest classifier

For all of the experiments in this work, we utilized a standard random forest classifier as our binary classifier[65] to analyze our training and testing sets, following the approach of Ghasemian et al.[13]. For Top-Sequential-Stacking, we train a random forest classifier on all topological predictors, and for Ensemble-Sequential-Stacking we train a random forest classifier with all topological features as well as additional predictors. Random Forest is an ensemble learning model that offers the flexibility to combine other predictors to enhance prediction results. In our case, we optimized the standard F measure to select the best parameters for Random Forest. The robustness of Random Forest allowed us to avoid over-fitting and gain insights into the significance of different features in our dataset through the Gini importance score. Admittedly, while Random Forest is a powerful and versatile classifier, it is not the only algorithm suited for the task at hand. Depending on the specifics of the problem and the dataset, other booster algorithms may provide better performance and accuracy. Therefore, it is important to carefully consider the strengths and weaknesses of different algorithms and select the one that is best suited for the particular situation.

### Assessment by average AUC scores

Our primary measure to evaluate link prediction performance is the area under the receiver operating characteristic curve (AUC), a standard measure used in these problems[11]. The AUC (Area Under the Curve) score is a scale-invariant and threshold invariant accuracy metric that is widely used in machine learning for evaluating and comparing the performance of binary classification models, such as link prediction. The AUC is defined mathematically as the probability that a uniformly random true positive case (missing link) is assigned a higher score than a uniformly random true negative case (non-edge), Pr(score(TP)) > Pr(score(TN)). It is conventionally calculated as the area under the receiver operating characteristic (ROC) curve, which plots the true positive rate against the false positive rate as a function of every classification threshold. AUC values range from 0.0 to 1.0, with a score of 0.5 corresponding to a random classifier and a score of 1.0 indicating a perfect classifier.[66] The AUC score is a useful metric because it takes into account the overall performance of a classifier across different threshold values, rather than just a single threshold. Additionally, AUC scores are a context-agnostic measure of the robustness of the method, while providing easy comparison with the current link prediction literature. All of the AUC scores reported in our results are averages over 50 runs, with 10 randomized repeats of the 5-fold cross-validation as described above. As noted previously, all AUC scores are computed for test dyads in the last layer of the corresponding dataset, sampled (as described above) to have equal-sized classes in the test set.

### Benchmarking methods and parameter choices

We compared three methods, Tensorial SBM, E-LSTM-D and Time-Series, using the default parameters provided by the original Github

repositories of the respective papers or methods. Tensorial-SBM[39] is a matrix (tensor) factorization model used for link prediction and community detection, which offers two models focused on node and edge grouping, respectively. In our study, we used the node-based Tensorial model with three default parameters: the number of node groups $K$, node pairs $J$, and layers $L$. The authors defaulted their parameters to be $K = J = L = 4$, while we kept $K = J = 4$ and varied the number of layers according to our experiment.

E-LSTM-D[29] is a deep learning method that predicts links on dynamical networks by utilizing long-term short memory (LSTM) and neural network. Their end-to-end encoder-decoder architecture automatically learns network representations, and the stacked LSTM module enhances the ability to learn temporal features. Their model has two default parameters: the number of units in the encoder and LSTM, set at 128 and 256, respectively. It is worth noting that although we used the same dataset *radoslaw*, we dissected it into 7 different layers, while the authors of E-LSTM-D chose to use only 2 layers, resulting in slightly different prediction outcomes for this dataset.

Time-Series[40] is a time-series auto-correlation over time method that utlize the predicted features from the past temporal snapshots of the network to conduct supervised learning for prediction of links. We adapted their method and used ARIMA with Random-Forest to predict each of the 41 individual features separately. For the completely unobserved case, the past temporal layers included are exactly equal to the parameter $u$, and for the partially observed case, we only use ARIMA on the test set to produce the prediction scores.

### Synthetic data

We use two synthetic temporal stochastic block models (T-SBMs) to mimic the real-world data and demonstrate the oracle performance of the method. In our T-SBM experiments, we generate multilayer networks each containing 200 nodes and 10 layers ($0 \leq t \leq 9$). Noting our $u = 6$ setting used throughout the present work and that we only test link prediction on the last layer (i.e., $t = L = 9$), the $0 \leq t \leq 2$ layers act to "burn in" the dynamics of the model, with the earliest data used for training the predictor coming from $G_3$ here.

The first model, which we identify as the "community-label T-SBM", is based on the Dirichlet distribution-based temporal stochastic block model presented in ref. 41 (see the reference for an extensive discussion about the model parameters therein). We highlight three model parameters that we vary: $p$ is the probability of simply copying a node label from one temporal layer to the next (otherwise, node labels are selected uniformly at random); $\mu$ is the fraction of edges added uniformly at random on top of the imposed community block structure; and $k$ is the number of communities. We consider $p \in \{0.9, 0.8, 0.7\}$, $\mu \in \{0.1, 0.2, 0.3\}$, and $k \in \{1, 2, 5, 10, 15\}$, generating a single temporal network realization for each parameter combination, yielding 45 different community-label T-SBM networks.

The second model, which we identify as the "edge-correlated T-SBM", is based on the edge-correlated degree-corrected temporal stochastic block model presented in[42], which we extend from two layers to multiple layers. Importantly, the community labels of the nodes do not change with time in this model. Instead, the correlation between different temporal layers is the edge correlation. Noting the full specification of the model is provided by the reference, we highlight the three parameters that we vary: $p$ is the edge correlation between one temporal layer and the next (i.e., the probability that an edge stays an edge) while $\mu$ and $k$ retain the same meanings as in the community-label T-SBM. By taking $p \in \{0.8, 0.7, 0.6\}$, $\mu \in \{0.1, 0.2, 0.3\}$, $k \in \{1, 2, 5, 10, 15\}$, we also create 45 different edge-correlated T-SBM synthetic networks.

### Reporting summary

Further information on research design is available in the Nature Portfolio Reporting Summary linked to this article.

## Data availability

The real world network data used in this study are publicly available as indicated in the SI in their corresponding database. All the synthetic network data are also publicly available at https://doi.org/10.5281/zenodo.10530764.

## Code availability

The code for implementing the method and reproducing the numerical experiments presented here can be found online at https://github.com/hexie1995/Sequential-Link-Prediction. Please refer to commit number 23740e9 on Nov 19, 2023, which also have a DOI at 10.5281/zenodo.10530764.

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

## Acknowledgements

We thank Marya Bazzi, Lucas Jeub, Roxana Pamfil, and Mason A. Porter for helpful discussions and conversation; and Khalique Newaz and Tijana Milenkovic for providing the bionet datasets. We are grateful for the use of the high performance computing clusters at the University of North Carolina at Chapel Hill (longleaf) and Dartmouth College (discovery7). A special thanks to Junyi Cheng[67] for helping with the graphical design of Fig. 1. Special thanks to Jonathan T. Lindbloom, Lizuo Liu, and Ryan Maguire for their help during the progress of this project. This work is supported in part by the Army Research Office under MURI award W911NF-18-1-0244 (X.H. and P.J.M.), the National Science Foundation under Grant No. 2030859 to the Computing Research Association for the CIFellows Project (A.G.), and the Learning & Academic research institution for Master's, PhD students, and Postdocs (LAMP) Program of the National Research Foundation of Korea (NRF) grant funded by the Ministry of Education (No. RS-2023-00301702) and the National Research Foundation of Korea (NRF) grant funded by the Korea government (MSIT) (No. RS-2022-00165916) (E.L.). The content is solely the responsibility of the authors and does not necessarily represent the official views of any agency supporting this research.

## Author contributions

X.H., A.G., and E.L. conceived and planned the experiments and contributed to the interpretation of the results. A.G. provided the base code from a previous paper[13]. A.G., A.C. and P.J.M. provided guidance on the direction of the research. X.H. took the lead in the numerical experiments and writing the initial manuscript draft, with P.J.M. supervising the project and revising the manuscript. All authors provided critical feedback and helped shape the research, analysis, and manuscript.

## Competing interests

The authors declare no competing interests.
