## [Peer Review File · Nature Communications]

Sequential Stacking Link Prediction Algorithms for Temporal NetworksReviewers' Comments:

Reviewer #1:

Remarks to the Author:

In this manuscript, He et al propose a stacking approach to link prediction in temporal networks. In their approach, features are obtained from static snapshots of the networks (which is computationally cheap) and then stacked to make temporal predictions, both on partially observed and completely unobserved network snapshots. This is compared to approaches in which (computationally costly) temporal generalizations of the static features are used.

Some of the results are interesting, including the general idea that stacking static features can lead to reasonably good predictions. However, despite my initial excitement with the results, I am afraid that none of the major claims in the abstract seem to hold beyond reasonable doubt:

- Stacking of static features is only shown to be superior to equivalent temporal features in specific situations.
- Stacking of static features is only proved to be quasi-optimal in some model networks but is clearly suboptimal on real networks.
- In general, stacking of static features alone is not better than some temporal models in the literature.

Considering this I think that the claims in the paper should be toned down and the paper would be appropriate for a more specialized journal with a lower standard for the novelty and ground-breaking nature of the reported results.

Below is a more nuanced discussion of my concerns.

The authors claim in the abstract that "many temporal topological features, in addition to having high computational cost, are less accurate in temporal link prediction than sequentially stacked static network features." Similar statements can be found in the introduction and discussion, such as "Comparing prediction results between several different temporally-extended features and sequentially stacking static features, we find that sequentially stacked features are at least as accurate, or far better, in temporal link prediction tasks while also being both substantially more computationally efficient and conceptually more well-defined." However, the results presented in the paper only support this claim for two very specific cases: 1) comparing stacked to temporal common neighbors (see next point), and 2) on T-SBM model networks (see below for problems with these models).

Regarding the comparison of stacked versus temporal common neighbors, the results do not seem very surprising, considering that the temporal version can be fully recovered from the stacked one, but not the other way around: "We define node j to be temporal common (sic?) neighbors of node i , if node j has been neighbors with node i in all of the q temporal layers prior to the target layer of interest." (I assume that 'common' is a mistake and this is the definition of neighbors, not common neighbors.) Thus, the temporal version loses information with respect to the stacked one.

For real networks, the non-stacking approaches Tensorial-SBM and E-LSTM-D often perform better than the temporally stacked approach. If I am not mistaken, in the partially observed setting the Tensorial-SBM performs best for 8 networks, E-LSTM-D for 7 and the stacking model for only 4. In the fully unobserved setting stacking is best 10 times and E-LSTM-D 9. Thus claims of the superiority of the stacking approach in real networks are not warranted. Of course Tensorial-SBM and E-LSTM-D can be added to the stacking model, but that somehow defeats the purpose of proving (and the claims) that stacking features across times is better than considering generalized multilayer/temporal features.

Regarding the stacking of methods, I find some results puzzling. For example for the chess network in fig 6a, Tensorial-SBM yields an almost perfect AUC and stacking yields $AUC > 0.7$, but stacking with

Tensorial-SBM yields $AUC < 0.6$. This does not look right.

Despite the fact that the stacking approach achieves near optimal performance on T-SBM experiments, that is not the case on real networks. In fact the performance of the stacking approach without multilayer methods (Tensorial-SBM and E-LSTM-D) is clearly suboptimal. This raises questions about claims such as this: "sequential stacking link prediction method uses 41 static network features that avoids detailed feature engineering choices and is capable of learning a highly accurate predictive distribution of future connections from historical data." Or this: "novel sequentially stacked link prediction algorithm for temporal networks that combines multiple static topological features within a temporally-aware meta-learning framework to produce nearly-optimal temporal link predictions." This also raises concerns about the suitability of the T-SBM models used for benchmarking, which seem to behave very differently to real networks.

Reviewer #2:

Remarks to the Author:

In this paper the authors tackle the problem of link prediction in temporal networks. Based on the information provided by the temporal network "up to time t ", they propose to study two sub-problems of link prediction:

[A] one where the task is to predict at time $t+1$ the existence of links in a target network which is partially observed (e.g., this amounts to a sort of missing-link paradigm in a network with only partial observation, based on the information stored by the network in the past.

The other sub-problem (B) is to completely predict the full target network at $t+1$, i.e. no partial observation of the network snapshot at $t+1$ is available.

The authors argue that to approach these problems, there are essentially two alternative methodological approaches: either

(i) extract from the temporal network some temporally-extended scalar topological features, or

(ii) to extract from each network snapshot a number of scalar network measures so as to build a feature vector for each snapshot, and stack together these vectors (for each network snapshot) to essentially construct a multivariate time series.

The authors propose a solution for (ii), and argue that it is computationally cheaper than (i), better well-defined, and provides equal or better performance. The choice of specific network features is based on previous experience by the authors in the task of link prediction in static networks.

Their validation predicates on using generative models of temporal networks (temporal stochastic block models), for which they are able to establish analytical bounds for the maximum possible performance (given the fact that these generative models are probabilistic), and they show their approach is "near-optimal" as compared to such bound. Results are compared against other prediction methods (Tensorial-SBM and E-LSTM-D).

Subsequently, they apply their framework in a range of empirical temporal networks spanning social, technological, transportation and biological systems, showing that their proposed method works well as compared to the benchmark. Finally, they explore importance scores and conclude that there are no "universal" topological features that are systematically more important for the prediction, something that they argue is aligned with the "no free lunch theorem".

The paper addresses an important question, of very general interest and applicability. The work is sound and the results impactful. The authors make a great effort to substantiate their findings, and

the work is indeed quite complete, encompassing a methodological framework, a sophisticated validation with generative models, and interesting applications to real data. Furthermore, the paper is very well written and reads smoothly.

I have a positive opinion on whether this paper deserves publication in Nature Communications, but I also have a number of comments, sources of confusion to be clarified and suggestions, that I believe the authors should address. I hope some of them will be of help to improve or clarify some aspects of this paper.

These comments are in assorted order of importance.

1) Confusion with learning algorithms

-- The paper doesn't explain well how the learning happens based on the sequentially stacked vectors, e.g. in [A] what is the $f()$ in $Y=f(X)$ where X is the sequentially stacked vector characterising a certain link and $Y=\{0,1\}$? I had presumed the authors are using for $f()$ some kind of binary classification algorithm, but this is not explained well, is $f()$ just a simple rule, a parametric algorithm which needs to be fitted (e.g. forecasting model), etc? The only allusion is given in Fig 1 where the authors say "we then use standard supervised learning algorithms..." These algorithms are not explicitly stated (or I couldn't find them)

-- Also, the authors only explain the case of "common neighbors", where the rule for $f()$ seems almost a trivial majority vote (a link will exist if it had existed, if I understood correctly). This rule is solely using persistence as the criterion, and therefore will be able to predict persistent links (or lack thereof), but not novelties, periodicities, or other sorts of fine grained temporal patterns. I was surprised to see (please correct me if I am wrong!!) that the authors don't train e.g. a discrete autoregressive process for each link, or something more sophisticated (but classic).

-- When the authors enrich the input data beyond common neighbors and add centrality measures and network latency, it is still not clear what algorithm do they use to predict existence of a link at time $t+1$ from the feature vector: is it a simple a priori rule? A binary classifier like a SVM, logistic regression, random forest, etc? Details are needed (I couldn't find them).

-- Similarly for [B].

-- The authors don't give a "minimal" explanation of what Tensorial-SBM and E-LSTM-D do. They refer to Methods and SI for details, but I couldn't find any detail besides pointing me to the respective papers. It would be good that the authors provide a concise summary of what these two methods amount to (specially, in terms of complexity, given that the authors benchmark theirs against these).

2) Framing

Other possible approaches (apparently not discussed in this work) for the more general problem [B] of network forecasting amounts either to extend classic time series forecasting models to networks, or alternatively to make use of graph neural networks (perhaps combined with recurrent architectures) [I believe some literature exists on the latter]. The authors don't discuss these alternatives in detail and present the narrative as if there is a dichotomy of choices, either (i) or (ii). I think it would be beneficial for the paper if the authors could put their approach and findings in that context, or at least acknowledge that (i) and (ii) are not the only possible approaches. Indeed, the authors benchmark their approach against one that uses recurrent networks, often with better results, but they don't pitch their message in these terms,

3) More on framing

To some extent, the narrative of the paper argues that using approach (i) is the "standard or default" and, accordingly, claims that approach (ii) should be preferred in terms of computational efficiency and performance. The authors cite references [31] and [32] to substantiate this. While interesting, they doesn't really seem to be manifesting a "standard approach" (also given the fact that in computer science the Graph Neural Networks touch on similar concepts, see my previous comment). In other words, the current writing feels a bit like stretching the narrative.

I wonder if the authors can either clarify whether this is a "false dichotomy" or to provide further evidence that approach (i) is the default.

4) Clarification of AUC

The main metric the authors use for prediction performance is the Area Under the Curve (AUC). I agree this is a good metric for the task at hand, and it is discussed to some degree in SI, but it is perhaps a bit obscure for the general reader. Considering Nat Comms has a large and diverse audience, it would be nice if the authors could provide a succinct clarification --for the sake of general readability-- of how this translates into something easily interpretable, i.e. what AUC=0.6 means? Relation to AUC and the confusion matrix (accuracy, false positive/negative), etc.

5) Results in empirical data

-- Page 17 last sentence, the authors say "the sequential algorithm performs as well as E-LSTM-D or better in all 19 networks in both partially observed and completely unobserved ..."

From figure 6, I don't think that's true? It's clearer in the right panel, where we see AUC is larger for E-LSTM-D than Sequential-Stacking for emaildnc, ant1, ant2 and maybe others.

-- AUC falls below 0.5 for some empirical networks. Does this mean that in those cases the algorithms behave worse than a random classifier?

-- How much variance is accumulated in the first two principal components? For "clusters" to be meaningful in the PC1/PC2 projection, I'd expect substantial. Also, the authors should give the definition of the Gini importance score.

6) Anticipating predictability

In general, I'd expect that TNs with lower predictability are those which are more randomly evolving. This could be easily checked (both in synthetic data and/or real data) by e.g. analysing the autocorrelation function of these temporal networks [Lacasa et al, Physical Review Research 4 (2022)] or their memory shape [Williams et al, Nature Communications 13 (2022)], and comparing it with e.g. AUC.

Lucas Lacasa

Reviewer #3:

Remarks to the Author:

The authors present a new approach to temporal link prediction in which features are stacked from multiple layers. The problem domain is predicting the links on layer t , using data from previous layers. The approach is to calculate static features from previous layers and stack them together. Previous work has used stacking for static link prediction, this work extends the previous work and applies it to the problem of temporal link prediction. Key results:

- Temporal stacking performs better and is computationally than temporally extended features. Tested on synthetic networks.
- Oracle level performance on synthetic networks where the underlying generative process is known.
- High performance on real-world network data.

The results indicate a robust and high performing method that can integrate many predictors and has good performance. The results on model performance are good (with a few concerns, see below), and the performance against simulated networks where optimal performance can be judged is strong. The claim that Temporal stacking is more computationally efficient is under supported, as they only test on a handful of temporally extended features.

There is a lack of detail on the methods, especially concerning the methods used for comparison (Tensorial-SBM and E-LSTM-D), which make it difficult to recreate the results they mention.

Some specific comments and questions:

- Thompson et. Al. (33) does not list betweenness centrality, degree centrality, or network latency in their list of temporally extended features (Table 1 in Thompson et. Al (33)). Can you indicate which temporally extended features you used, and if not defined in Thompson et. Al (33), please include definitions within this section?
 - I am very surprised at the computational cost of the temporally extended features you mention on page 10. Can you compare the networks you tested against with the ones Thompson et. Al. (33) applied to, since it seems the temporally extended measures were applied there? Were the networks you were considering more complex than the ones in Thompson?
- Which version of the Tensorial-SBM did you use? Tarres et. al include two types (T-MBM and B-MBM) in their article, and note that the models differ on what types of networks they perform best on.
 - Additionally, what were the parameters used in the Tensorial-SBM?
- Could you provide a table with the AUC value on the real world data sets?
- Can you elaborate on the parameters used within the comparison method E-LSTM-D? I note that in the original E-LSTM-D paper (Chen et. Al), the AUC results for the RADOSLAW network are 0.98 and 0.98, whereas in your paper the results show approximately 0.88 (once again, a table with the AUC values per real world data set and method would be useful).

Dear Reviewers,

We would like to express our gratitude for your valuable comments, which have greatly helped us enhance the quality of our paper. In response to your feedback, we have made significant revisions to the manuscript. The major additions are indicated with **blue text** for ease of identification in the revised version (while text that was moved around but otherwise without major change is not highlighted with color).

In the detailed response below we have broken out distinct comments and suggestions, and responded point-by-point to describe how we addressed them to improve the overall clarity and value of the paper.

As a summary, the major changes we made include:

- **Contributions:** We have more explicitly highlighted the contributions of our paper, which include the introduction of the topological stacking framework (Top-Sequential-Stacking) for link prediction in temporal networks and the associated development of an ensemble-learning approach (Ensemble-Sequential-Stacking) that incorporates the topological stacking algorithm along with other suitable predictors. We emphasize that our work not only avoids the complex definitions and costly calculations of temporal topological features but also introduces a framework to enhance overall temporal link prediction accuracy.
- **Comparison with Time-Series Features:** We have addressed the reviewer-raised concern regarding the limited comparison between sequentially stacked features and temporal topological features. In response, we have conducted new experiments using the time-series auto-regressive ARIMA model as a benchmark. We compare all 41 sequentially stacked individual predictors against the time-series features and demonstrate that the sequentially stacked predictors outperform in terms of both accuracy and time-efficiency. This additional experiment strengthens the evidence supporting our results and addresses the issue of limited comparisons with temporal topological features.
- **Temporal Autocorrelation Analysis:** We have included a new experiment that calculates the temporal autocorrelation function for all the networks studied in our research. The results of this analysis, demonstrating the predictability of the networks, are now presented in the results section and Supplementary Information (SI).
- **Code Issue:** During the course of our new experiments, we encountered a minor issue caused by a change in the updated Python pandas version. We have promptly addressed this issue in our code and have provided the corrected results. Crucially, the overall results and our conclusions remain unchanged, but we wanted to be up-front about this issue.
- **Additional Details:** We have addressed the missing information and experiment settings of the benchmarking methods in the manuscript. We have also provided further details about the metrics used, such as AUC and Gini importance. We apologize for the oversight in the initial version and appreciate the reviewers for raising these points.

We are grateful to you for your valuable feedback and we believe these changes greatly strengthen the overall clarity and comprehensiveness of our manuscript, significantly improving the paper. We hope you will now find it suitable for publication in *Nature Communications*.

Best regards,

Xie He, Amir Ghasemian, Eun Lee, Aaron Clauset, Peter Mucha

Review 1

R1.1 *In this manuscript, He et al propose a stacking approach to link prediction in temporal networks. In their approach, features are obtained from static snapshots of the networks (which is computationally cheap) and then stacked to make temporal predictions, both on partially observed and completely unobserved network snapshots. This is compared to approaches in which (computationally costly) temporal generalizations of the static features are used.*

Some of the results are interesting, including the general idea that stacking static features can lead to reasonably good predictions. However, despite my initial excitement with the results, I am afraid that none of the major claims in the abstract seem to hold beyond reasonable doubt

...[distinct points from reviewer 1 separated out into R1.2, R1.3 and R1.4 below]...

Considering this I think that the claims in the paper should be toned down and the paper would be appropriate for a more specialized journal with a lower standard for the novelty and ground-breaking nature of the reported results.

Our response: We appreciate the constructive criticism from the reviewer’s feedback, which has allowed us to clarify our contributions and present compelling evidence of the effectiveness of our approach. We believe the revised manuscript addresses these concerns raised and provides a more robust and convincing presentation of our work. In particular, we acknowledge the previous version of the manuscript was not sufficiently clear on some of these points, and we appreciate the opportunity to address and alleviate some of the confusion of the original manuscript. We have made significant improvements to clarify our contributions and highlight the two main aspects of our work: the stacking of static features as a replacement for temporal features and the development of a meta-learning framework for integrating other approaches (highlighted in blue on pages 3, 4, 5 & 6). We hope these clarifications alleviate any concerns and convince the reviewer of the value and significance of our contributions. Detailed revisions of the manuscript could be found in the response below.

R1.2 - *Stacking of static features is only shown to be superior to equivalent temporal features in specific situations.*

Our response: We acknowledge the concern regarding the limited number of features considered in our initial comparison. We appreciate the suggestion to explore additional features and we are open to considering them. As we mentioned in the first draft, the challenge lies in the lack of well-defined temporal network features. To address this concern comprehensively, we have performed additional experiments using ARIMA models to generate time series features for each of the 41 individual network features we have used in our method. We compared the performance of these time series features against the sequentially stacked static features on all real-world and synthetic networks. The results of this comparison can be found in the Results section (Page 14, Figure 3) of the revised manuscript. Additionally, a detailed analysis of each feature and dataset is presented in the Supplementary Information (Section F). By including this comparison and demonstrating that sequentially stacked features achieve better accuracy and time complexity than time series features, we

believe we have provided much stronger evidence to support our claims. We hope these additional experiments address this concern expressed by the reviewer.

R1.3 - *Stacking of static features is only proved to be quasi-optimal in some model networks but is clearly suboptimal on real networks.*

Our response: In the revised manuscript, we provide clearer evidence of the effectiveness of our proposed methods on real world networks. As shown in the Results section (Page 21, Figure 6) — corrected in our code to account for use of a different pandas version — the Top-Sequential-Stacking method alone outperforms other state-of-the-art methods in most cases. In addition to the better performance on the two different temporal SBMs, Top-Sequential-Stacking alone consistently outperforms all other individual predictors in 17 out of 19 cases for the partially observed target layers for the real networks, and in 12 out of 19 cases for the completely unobserved target layers. Additionally, when combining sequential stacking with all other individual predictors in Ensemble-Sequential-Stacking, we achieve the best AUC scores in 17 of the 19 real-world networks for the partially observed case and 16 out of 19 cases for the completely unobserved target layers. For the cases where Top-Sequential-Stacking or Ensemble-Sequential-Stacking is not doing the best, they are at most only 0.03 below the best performer.

R1.4 - *In general, stacking of static features alone is not better than some temporal models in the literature.*

Our response: We appreciate the opportunity to further clarify our findings and address the reviewer’s concerns. In the updated results, we have demonstrated that Top-Sequential-Stacking alone outperforms most other individual predictors, including temporal topological features and time-series extended features, in terms of both accuracy and computational efficiency (see also R1.2 and R1.3). This provides strong evidence of the effectiveness of our designed stacking algorithm. Additionally, we emphasize that the underlying nature of Top-Sequential-Stacking framework extends naturally to use and obtain even higher accuracy as an ensemble learning method. The purpose of using ensemble learning is to combine multiple predictors and leverage their strengths to improve overall prediction performance. In our ensemble learning framework, the added model-based techniques are not competitors but rather complementary components that contribute to the collective prediction accuracy. As shown in Figures 4 (Page 17), 5 (Page 18), and 6 (Page 21), the performance of Ensemble-Sequential-Stacking, which incorporates Top-Sequential-Stacking and other individual predictors, consistently outperforms each predictor separately. Therefore, both Top-Sequential-Stacking and Ensemble-Sequential-Stacking taken together are important contributions of our work. Top-Sequential-Stacking addresses the need for interpretable features by replacing temporal topological features, while Ensemble-Sequential-Stacking provides a powerful framework for integrating multiple predictors and achieving superior prediction performance. These two aspects of our work are complementary and work together to enhance the accuracy and interpretability of temporal link prediction.

R1.5 *Below is a more nuanced discussion of my concerns. The authors claim in the abstract that “many temporal topological features, in addition to having high computational cost, are less*

accurate in temporal link prediction than sequentially stacked static network features.” Similar statements can be found in the introduction and discussion, such as “Comparing prediction results between several different temporally-extended features and sequentially stacking static features, we find that sequentially stacked features are at least as accurate, or far better, in temporal link prediction tasks while also being both substantially more computationally efficient and conceptually more well-defined.” However, the results presented in the paper only support this claim for two very specific cases: 1) comparing stacked to temporal common neighbors (see next point), and 2) on T-SBM model networks (see below for problems with these models).”

Our response: We appreciate the reviewer’s comments regarding the comparison of features. We apologize for the misunderstanding due to our wording in the original version of the draft. In the original version of the manuscript, we already compared beyond common neighbors to also consider four other features for benchmarking: betweenness centrality, closeness centrality, degree centrality, and network latency. We acknowledge that many temporal network features are either not well-defined or challenging to compute, which is why we selected these five features for comparison and discussion on T-SBMs. However, we understand the importance of a comprehensive analysis and have now, in response to comments including this one, included time-series features for comparison on all datasets and all features (as mentioned in our response to R1.2)

R1.6 *Regarding the comparison of stacked versus temporal common neighbors, the results do not seem very surprising, considering that the temporal version can be fully recovered from the stacked one, but not the other way around: “We define node j to be temporal common (sic?) neighbors of node i , if node j has been neighbors with node i in all of the q temporal layers prior to the target layer of interest.” (I assume that ‘common’ is a mistake and this is the definition of neighbors, not common neighbors.) Thus, the temporal version loses information with respect to the stacked one.*

Our response: We appreciate the reviewer’s comment regarding the definition of temporal common neighbors. We would like to clarify that the use of temporal common neighbors in our approach is intentional and serves as a foundation for introducing more sophisticated and complex features. Indeed, the words we used in the original manuscript were “illustrative example” and we raised precisely this same point about how the temporal version of common neighbors has less information than sequentially stacking static common neighbors: “Noting that this temporal common neighbors definition ignores all variation across sequential layer...” Nevertheless, we started simply with common neighbors (again, as an “illustrative example”) because of its prior use in temporal network analysis and link prediction. The more advanced features (see also R1.5), which are discussed in detail in the later part of the results section (and Pages 17, 18, 21 and Figures 4, 5 & 6), allow us to capture a deeper understanding of the temporal dynamics and relationships in the network. By incorporating comparisons with these features, we aim to enhance the performance and effectiveness of our proposed method in temporal link prediction.

R1.7 *For real networks, the non-stacking approaches Tensorial-SBM and E-LSTM-D often perform better than the temporally stacked approach. If I am not mistaken, in the partially observed setting the Tensorial-SBM performs best for 8 networks, E-LSTM-D for 7 and the stacking model for only 4. In the fully unobserved setting stacking is best 10 times and E-LSTM-D 9. Thus*

claims of the superiority of the stacking approach in real networks are not warranted. Of course Tensorial-SBM and E-LSTM-D can be added to the stacking model, but that somehow defeats the purpose of proving (and the claims) that stacking features across times is better than considering generalized multilayer/temporal features.

Our response: Please see our answer to R1.3 above. Additionally, we would like to re-emphasize that while the sequentially stacked features on their own frequently provide very accurate link prediction results, we do not need to surpass the performance of generalized multilayer methods but rather naturally leverage their strengths within our ensemble-learning framework. As demonstrated in the original version of our work and reiterated in our cover letter, R1.1 and R1.2, sequentially stacked static features outperformed temporal features in terms of both performance and time complexity in most cases. In the revised version, we have further provided evidence that stacked static features perform comparably or even better than time-series features, while also offering lower time complexity. Therefore, our approach offers a promising alternative that combines the benefits of all of these otherwise seemingly disparate approaches: stacked static features, time-series features, and general multilayer methods.

R1.8 *Regarding the stacking of methods, I find some results puzzling. For example for the chess network in fig 6a, Tensorial-SBM yields an almost perfect AUC and stacking yields $AUC > 0.7$, but stacking with Tensorial-SBM yields $AUC < 0.6$. This does not look right.*

Our response: We extend our sincere gratitude to the reviewer for their invaluable comments. In the revised draft we have addressed the concern about the AUC score, which appears to be attributed to overfitting and the pandas version used there. We have rectified this issue and carefully re-run the code to ensure the accuracy and reliability of our results (see Figure 6 on Page 21). Thank you for bringing this to our attention, and we appreciate your thorough review.

R1.9 *Despite the fact that the stacking approach achieves near optimal performance on T-SBM experiments, that is not the case on real networks. In fact the performance of the stacking approach without multilayer methods (Tensorial-SBM and E-LSTM-D) is clearly suboptimal. This raises questions about claims such as this: “sequential stacking link prediction method uses 41 static network features that avoids detailed feature engineering choices and is capable of learning a highly accurate predictive distribution of future connections from historical data.” Or this: “novel sequentially stacked link prediction algorithm for temporal networks that combines multiple static topological features within a temporally-aware meta-learning framework to produce nearly-optimal temporal link predictions.” This also raises concerns about the suitability of the T-SBM models used for benchmarking, which seem to behave very differently to real networks.*

Our response: We appreciate the reviewer’s feedback, which we have endeavored to address in the revised version. We have addressed the issue regarding real-world network performance in R1.2 and R1.3, and specifically in Figure 6 on page 21 in the revised draft, which we believe provides compelling evidence to support our claims regarding the optimality of the stacking technique in the context of real networks. Additionally, we would like to emphasize that the T-SBMs have proven to be effective in capturing qualitatively different behaviors in performance that are then similar to that seen in the real-world networks, as demonstrated by the common-neighbor and

other temporal feature comparisons in the original manuscript. The known structure of these models also provides fundamentally essential insights in calculating the oracle-level performance, while also enhancing our understanding of the performance of our proposed approach. Their effectiveness in capturing the real-world network features are also shown in the original papers for them. Of course, we agree with the reviewer that “no model is a good model” and use these two T-SBMs cautiously only to present that they resemble some part of the behavior on the real-world networks. This is also the precise reason why we have used 19 real-world networks in the experiment.

Review 2

R2.1 *The paper addresses an important question, of very general interest and applicability. The work is sound and the results impactful. The authors make a great effort to substantiate their findings, and the work is indeed quite complete, encompassing a methodological framework, a sophisticated validation with generative models, and interesting applications to real data. Furthermore, the paper is very well written and reads smoothly.*

I have a positive opinion on whether this paper deserves publication in Nature Communications, but I also have a number of comments, sources of confusion to be clarified and suggestions, that I believe the authors should address. I hope some of them will be of help to improve or clarify some aspects of this paper.

Our response: We would like to express our sincere gratitude to the reviewer for their valuable and constructive comments. We genuinely appreciate the time and effort they have dedicated to reviewing our manuscript. Their suggestions and feedback have been instrumental in enhancing the quality of our work, and we are grateful for their assistance in improving the draft.

R2.2 *1) Confusion with learning algorithms – The paper doesn't explain well how the learning happens based on the sequentially stacked vectors, e.g. in [A] what is the $f()$ in $Y=f(X)$ where X is the sequentially stacked vector characterising a certain link and $Y=0,1$? I had presumed the authors are using for $f()$ some kind of binary classification algorithm, but this is not explained well, is $f()$ just a simple rule, a parametric algorithm which needs to be fitted (e.g. forecasting model), etc? The only allusion is given in Fig 1 where the authors say “we then use standard supervised learning algorithms...” These algorithms are not explicitly stated (or I couldn't find them)*

Our response: We apologize for the oversight in not providing the implementation details in the initial version of the manuscript. We have now included the implementation details of the random forest algorithm in the Methods section. These details can be found on page 32 of the revised manuscript. Thank you for bringing this to our attention, and we appreciate your understanding.

R2.3 *Also, the authors only explain the case of “common neighbors”, where the rule for $f()$ seems almost a trivial majority vote (a link will exist if it had existed, if I understood correctly). This rule is solely using persistence as the criterion, and therefore will be able to predict persistent links (or lack thereof), but not novelties, periodicities, or other sorts of fine grained temporal patterns. I was surprised to see (please correct me if I am wrong!!) that the authors don't train e.g. a discrete autoregressive process for each link, or something more sophisticated (but classic).*

Our response: We thank the Reviewer for this comment because it motivated us to provide further comparisons, specifically by incorporating autoregressive time series analysis (ARIMA) as a comparison metric. In the revised manuscript, we provide a comprehensive comparison of the sequentially stacked features against the autoregressive features. The results of this comparison can be found in the Results section of the manuscript (Page 14, 17, 18, 21 and Figures 3–6.). Additionally, for more detailed information, we have included Section F in the Supplementary Information, which provides additional insights and analysis on the autoregressive

features and their performance. Thank you for bringing this to our attention, and we appreciate your feedback.

R2.4 – *When the authors enrich the input data beyond common neighbors and add centrality measures and network latency, it is still not clear what algorithm do they use to predict existence of a link at time $t+1$ from the feature vector: is it a simple a priori rule? A binary classifier like a SVM, logistic regression, random forest, etc? Details are needed (I couldn't find them).– Similarly for [B]. – The authors don't give a “minimal” explanation of what Tensorial-SBM and E-LSTM-D do. They refer to Methods and SI for details, but I couldn't find any detail besides pointing me to the respective papers. It would be good that the authors provide a concise summary of what these two methods amount to (specially, in terms of complexity, given that the authors benchmark theirs against these).*

Our response: We apologize for the previous oversight. In the revised version of the manuscript, these details can be found in the Methods section of the paper, specifically on page 32. They are all conducted with random forests. Additionally, we have provided comprehensive information highlighted on the implementation details of all benchmarking methods in the revised Methods section on page 33. We appreciate your feedback and thank you for bringing this to our attention.

R2.5 2) *Framing Other possible approaches (apparently not discussed in this work) for the more general problem [B] of network forecasting amounts either to extend classic time series forecasting models to networks, or alternatively to make use of graph neural networks (perhaps combined with recurrent architectures) [I believe some literature exists on the latter]. The authors don't discuss these alternatives in detail and present the narrative as if there is a dichotomy of choices, either (i) or (ii). I think it would be beneficial for the paper if the authors could put their approach and findings in that context, or at least acknowledge that (i) and (ii) are not the only possible approaches. Indeed, the authors benchmark their approach against one that uses recurrent networks, often with better results, but they don't pitch their message in these terms,*

3) *More on framing To some extent, the narrative of the paper argues that using approach (i) is the “standard or default” and, accordingly, claims that approach (ii) should be preferred in terms of computational efficiency and performance. The authors cite references [31] and [32] to substantiate this. While interesting, they doesn't really seem to be manifesting a “standard approach” (also given the fact that in computer science the Graph Neural Networks touch on similar concepts, see my previous comment). In other words, the current writing feels a bit like stretching the narrative. I wonder if the authors can either clarify whether this is a “false dichotomy” or to provide further evidence that approach (i) is the default.*

Our response: We greatly appreciate your feedback. In response to your comment, we have made several significant changes to the manuscript. Firstly, we have expanded the discussion on the importance of stacking static features as a replacement for temporal features in the Introduction section. This shift in emphasis highlights the unique contribution of our approach in leveraging static features to improve temporal link prediction (pages 3–6). Furthermore, we have included a detailed discussion on time series methods in the benchmarking section, specifically in the comparison with our sequentially stacked static features. This discussion can be found in the Results section, specifically on Page 14, 17, 18, 21, and Figures 3–6. We have carefully restructured the Introduction, rewording and reorganizing it to better emphasize the utilization of stacking static features as a replacement for temporal features and

the implementation of our ensemble learning stacking approach as a near optimal solution. We hope that these changes address your concerns and provide a more comprehensive and coherent discussion of our work. Thank you for your valuable feedback.

R2.6 4) *Clarification of AUC* The main metric the authors use for prediction performance is the Area Under the Curve (AUC). I agree this is a good metric for the task at hand, and it is discussed to some degree in SI, but it is perhaps a bit obscure for the general reader. Considering Nat Comms has a large and diverse audience, it would be nice if the authors could provide a succinct clarification –for the sake of general readability– of how this translates into something easily interpretable, i.e. what AUC=0.6 means? Relation to AUC and the confusion matrix (accuracy, false positive/negative), etc.

Our response: The details of AUC scores are now included in the Methods section on page 32. We appreciate your attention to detail and apologize for any oversight in our previous response.

R2.7 5) *Results in empirical data* – Page 17 last sentence, the authors say “the sequential algorithm performs as well as E-LSTM-D or better in all 19 networks in both partially observed and completely unobserved ...” From figure 6, I don’t think that’s true? It’s clearer in the right panel, where we see AUC is larger for E-LSTM-D than Sequential-Stacking for emaildnc, ant1, ant2 and maybe others. – AUC falls below 0.5 for some empirical networks. Does this mean that in those cases the algorithms behave worse than a random classifier?

Our response: Thank you for spotting this. We acknowledge there was a bug in the original code with regard to its use of a previous pandas version that has since been identified and rectified. We have updated the figures in the Results section on Page 21, Figure 6 (and also Pages 17–18, Figures 4–5).

R2.8 – *How much variance is accumulated in the first two principal components? For “clusters” to be meaningful in the PC1/PC2 projection, I’d expect substantial. Also, the authors should give the definition of the Gini importance score.*

Our response: Thank you for your comment. We have updated the Figure 7 on page 23 to include that information. The definition of the Gini importance are also now included on page 23, highlighted in blue.

R2.9 6) *Anticipating predictability* In general, I’d expect that TNs with lower predictability are those which are more randomly evolving. This could be easily checked (both in synthetic data and/or real data) by e.g. analysing the autocorrelation function of these temporal networks [Lacasa et al, *Physical Review Research* 4 (2022)] or their memory shape [Williams et al, *Nature Communications* 13 (2022)], and comparing it with e.g. AUC.

Our response: We greatly appreciate the reviewer’s comments. This is now included in the results section for discussion on predictability in general (specifically highlighted on page 19 and 22) and detailed scores are provided in Section E of the SI.

Review 3

R3.1

The results indicate a robust and high performing method that can integrate many predictors and has good performance. The results on model performance are good (with a few concerns, see below), and the performance against simulated networks where optimal performance can be judged is strong. The claim that Temporal stacking is more computationally efficient is under supported, as they only test on a handful of temporally extended features.”

Our response: We are grateful for the reviewer’s comments and feedback, and we have taken them into consideration in revising the main text and accompanying files. As requested, we have made the necessary changes to provide additional support for our claim. Specifically, we have included a performance and time comparison for all features, along with the results of the auto-correlated time-series analysis. These updates can be found throughout the results section of the main text, specifically on Pages 14, 17, 18, 21, and Figures 3–6, as well as in Section F of the SI.

R3.2 *“There is a lack of detail on the methods, especially concerning the methods used for comparison (Tensorial-SBM and E-LSTM-D), which make it difficult to recreate the results they mention.”*

Our response: We would like to express our gratitude for the reviewer’s concern and helpful suggestion. In response, we have added a comprehensive section in the Methods section specifically dedicated to describing the details and implementation of the other benchmarking methods. We believe that this addition will greatly facilitate the replication process and enhance the reproducibility of our study. The new section can be found on Page 33, and further information and discussion related to benchmarking methods can be found throughout the results and methods sections.

R3.3 *Thompson et. Al. (33) does not list betweenness centrality, degree centrality, or network latency in their list of temporally extended features (Table 1 in Thompson et. Al (33)). Can you indicate which temporally extended features you used, and if not defined in Thompson et. Al (33), please include definitions within this section?*

Our response: We appreciate the reviewer for pointing out the differences in the feature names. In response to this, we have included the alternative names of the features mentioned by Thompson et al. highlighted on Page 13. Additionally, we have used the definition of betweenness centrality from Zao et al. as indicated in the main text also highlighted on Page 13. Thank you for bringing this to our attention and helping us improve the accuracy of our manuscript.

R3.4 *I am very surprised at the computational cost of the temporally extended features you mention on page 10. Can you compare the networks you tested against with the ones Thompson et. Al. (33) applied to, since it seems the temporally extended measures were applied there? Were the networks you were considering more complex than the ones in Thompson?*

Our response: Indeed, the networks utilized in Thompson et al.’s study consist of subgraphs with only 26 nodes each which are dissected from a 264 node graph, as

mentioned in their work. In contrast, our smallest networks are twice as large (50 nodes). Unfortunately, Thompson et al. did not fully specify edge densities used in their networks, nor did they specify the number of consecutive temporal layers they considered for calculating their temporal features. We assume that disparities in network size, edge density, and numbers of temporal layers cause the computational requirements for our analysis to be significantly higher, resulting in longer computation times. We have included this clarification in the manuscript to underscore the distinctions between the two studies on Page 13.

R3.5 *Which version of the Tensorial-SBM did you use? Tarres et. al include two types (T-MBM and B-MBM) in their article, and note that the models differ on what types of networks they perform best on. Additionally, what were the parameters used in the Tensorial-SBM?*

Our response: We appreciate the feedback and have made the necessary updates to the manuscript. We have used T-MBM (Temporal Multilayer Block Model) and the methods section now includes detailed information about the T-MBM and its implementation. The parameter choices for T-MBM are based on the default settings from the original paper, with some modifications made to the number of layers. The specific details about the parameters and implementations can be found on Pages 33 and 34.

R3.6 *Could you provide a table with the AUC value on the real world data sets?*

Our response:

We appreciate the comment and have included the necessary information in Section D of the SI.

R3.7 *Can you elaborate on the parameters used within the comparison method E-LSTM-D? I note that in the original E-LSTM-D paper (Chen et. Al), the AUC results for the RADOSLAW network are 0.98 and 0.98, whereas in your paper the results show approximately 0.88 (once again, a table with the AUC values per real world data set and method would be useful).*

Our response: We appreciate the opportunity to provide clarification about this. We have now included the relevant information in the methods section on Pages 32–34, providing details about the parameters and implementations. In particular, it is important to note the differences in the dataset separation strategy, with the RADOSLAW dataset being separated into two layers in the E-LSTM-D paper and into seven layers in our study (with only five layers used for training), and that this difference may contribute to slight variations in the performance outcomes between the two approaches.

Reviewers' Comments:

Reviewer #1:

Remarks to the Author:

N/A

Reviewer #2:

Remarks to the Author:

I appreciate the authors' efforts on addressing all reviewers' comments and I enjoyed very much reading the revised manuscript. I endorse the publication of this nice piece of work.

L.L.

Reviewer #3:

None

Dear Reviewers,

We extend our sincere gratitude to you for your invaluable comments. In the detailed response provided below, we have addressed each of the distinct comments and concerns in depth. To sum up, we feel there are mainly two points that needed further clarification that we have aimed to address in the present response:

- *The concern regarding the difference in the plotting of Figure 6 from the originally submitted version to that in the first revision:* This difference turns out to be (somewhat embarrassingly for us) due to the first author mistakenly loading old files from January 2022's results multiple times in the process of creating figure 6 in the original submission. Indeed, upon closer inspection of the original version's figure 6, one can observe that part of the completely unobserved case exhibits the exact same shape as appears in the partially observed case, which results from the coding error that reused the same (incorrect) file in that very same plotting code. To substantiate this explanation, we have provided screenshots as evidence, along with the original submission's plotting code and files, all of which confirm this mistake. We recognize that this is an incredibly unfortunate oversight on our part, and we want to underscore that we have double checked that the plotting in the revised manuscript is indeed accurate and free from these errors. Moreover, we wish to highlight that the repository with all the code and data necessary to perform our feature stacking method has been referenced and publicly available online since the original version. Additionally, motivated both by the most recent round of reviews and the detective work we did to chase down our bug in the plotting routine of the original manuscript and by a desire to more greatly improve transparency, we have now further updated the repository to include all the codes necessary to use the previously existing packages (by other authors) for the full "Ensemble-Sequential-Stacking" method that incorporates the sequential stacking together with other suitable predictors. We wholeheartedly invite the reviewers to download and run the codes on both the synthetic and real-world networks if they need to do so to dispel any remaining doubts or concerns.
- *The use of the word "optimal" and any claims about which method is better in one setting vs. another:* Aiming to best and most constructively improve the paper from Reviewer 1's careful comments, we have revised the paper considering every point where the word "optimal" appears in the previous revision and changed as needed to align with the simple facts as substantiated by our results but nothing further. All of these changes are highlighted by blue text in the newly revised manuscript (along with all remaining instances of "optimal" in the main text). (The only appearance of the word in the supplement was where we "calculate the theoretically optimal level of link predictability in our synthetic temporal networks," so we have left that as is.)

Last but not least, as the first author (who made the plotting code mistake), I would like to personally thank all the reviewers for their help in shaping this paper and for providing guidance along the way. Your careful reviews have greatly helped me avoid an accidental yet nevertheless serious academic integrity issue and debugged the problem I overlooked. This is a valuable lesson about version control and carefulness that I will never forget. Thank you!

We sincerely hope that the Reviewers recognize that we have embraced their comments with a constructive spirit and that they now support publication of the manuscript.

Best regards,
Xie He, on behalf of all authors

Review 1

R1.1

I have thoroughly reviewed the revised manuscript and accompanying rebuttal letter. Regrettably, the revised version introduces more questions than it answers.

In my initial assessment, I highlighted the paper's unsubstantiated primary claims. Specifically, the paper asserted the superiority and quasi-optimality of the stacking approach over multilayer methods (Tensorial-SBM and E-LSTM-D). However, as shown in the original submission's Figure 6, out of the 19 real networks analyzed, multilayer approaches demonstrated greater accuracy than stacking for 15 of those networks. I've included the original Figure 6 below for your reference (focusing on the partially observed case on the left; similar observations apply to the completely unobserved case on the right):

To illustrate, consider the example of the chess network: Tensorial-SBM exhibited an AUC near 1, sequential-stacking around 0.7, and E-LSTM-D just above 0.5. Similar trends were apparent for other networks such as *obrazil* and *bionet1*.

We are grateful to the Reviewer for their perseverance on this topic, as it has caused us to do a more careful “detective work” search of all of our files to try to explain this answer.

In short: there was a coding bug in the function that **plotted** this figure in the original manuscript. The bug was simple in that it loaded a collection of data files storing computational results, but the files it loaded were very old files from the very early days of this project. We have documented all of this in detail below at the end of this response letter, if you want to see it. We are quite embarrassed by this mistake, and that we didn't catch it sooner, and we are grateful that the Reviewer forced us to chase it down.

This bug did **not** propagate into the revised version of the manuscript because we added other methods and that changed the combinations that we included and we re-ran everything, so it was a completely new block of code to generate the new figure.

The factual claim made based on the correct plot in the revised manuscript is as follows: “We observe that, between them, Top-Sequential-Stacking and Ensemble-Sequential-Stacking achieves the highest accuracy on 16 of 19 real-world datasets in the completely-unobserved setting and on 17 of 19 in the partially-observed setting and in the remaining cases we observe only minimal reduction in performance relative to the best predictor.” We hope the Reviewer agrees that this straightforward statement is accurate.

Further acknowledging the Reviewer’s concern about the use of the word “optimal”, we have revised the paper carefully checking every use of the word and modified as needed and appropriately to language that we hope the Reviewer can agree with. We note that in some cases we did still think the word was appropriate, especially insofar as times we used it in a way separate from making any claim of our results. To try to help the Reviewer efficiently check this, we have highlighted all of these changes in blue text along with all occurrences of the word “optimal” even if it isn’t a change from the previous version. We have also changed the GitHub repository name for good measure. If these modifications are not sufficient, we hope the Reviewer can be concrete about any additional desired edits.

R1.2

Now, turning to the revised version and the rebuttal letter, the outstanding issues are the following:

1. The authors attribute the previously noted concern to a coding bug related to the version of Pandas used for analysis. Yet, they fail to provide specifics about this serious error in the original submission. Furthermore, the rebuttal letter does not clarify why the initial claim of stacking’s superiority was made despite multilayer methods having (at the time, at least) higher accuracy for most empirical networks. From the available information (two manuscript versions plus rebuttal), it appears as though the authors were driven to a predetermined conclusion, irrespective of the empirical evidence.

There are two important points here that we want to respond to.

Yes, we did indeed previously — and erroneously, it turns out — attribute the change in results to a coding bug related to the version of Pandas used. As can be clearly and easily seen by the history available in the repository for our project that has been available since the beginning, this was the only substantial change to the actual sequential stacking method. We had one set of results in the original manuscript (which we now know was from plotting data from wrong, very old files), we changed only this one meaningful part of the code, and then we had different results. Was it perhaps lazy of us to assume that if we only changed X that X must be the cause for the changes? There are some changes due to the Pandas version change that are important and that are fixed, but that’s nothing compared to the root cause of the difference here: the bug in our plotting code. It was definitely incorrect of us to make this leap. Again, the error in our logic was that we didn’t account for the possibility (indeed, the fact, as it turned out) that there was an error in the plotting routine we wrote for this figure in the original manuscript. Again, all of this is documented below if you want to see it.

On to the second point that accuses us of being “driven to a predetermined conclusion,” we merely reiterate the results as previously stated for the correctly-plotted figure in the revision: “We observe that, between them, Top-Sequential-Stacking and Ensemble-Sequential-Stacking achieves the highest accuracy on 16 of 19 real-world datasets in the completely-unobserved setting and on 17 of 19 in the partially-observed setting and in the remaining cases we observe only minimal reduction in performance relative to the best predictor.” We hope that the Reviewer can agree with this claim as stated. If not, we would welcome concrete edits about what is not okay with this statement.

R1.3

2. In any case, after presumably correcting the unspecified error, the new results are documented in the updated Figure 6, presented below:

Strikingly, **these new results bear absolutely no resemblance to the original ones**. For instance, in the case of the *chess* network, stacking approaches now exhibit an AUC nearing 1 (previously 0.7), while Tensorial-SBM and E-LSTM-D achieve approximately 0.75 (formerly 1 and 0.5, respectively). Notably, the Tensorial-SBM's AUC for the *obrazil* network drops from nearly 0.95 in the initial submission to less than 0.5—shifting **from near perfection to a performance worse than random guessing**.

I cannot conceive of a bug capable of transforming an algorithm from near randomness to near perfection. In my experience, coding errors introduce noise that uniformly degrades results across experiments. Hence, it remains inexplicable why the new outcomes bear no correlation whatsoever with the original ones—I find it difficult to arrive at a plausible explanation for this incongruity.

First and foremost, I, as the first author and the primary contributor to the coding of the experiments, wish to offer my sincere apologies and appreciation. I totally understand the frustration and doubt from the reviewer's point of view. In fact, I genuinely appreciate the reviewer's comment because this helped me find out what was really happening in these codes.

In my plotting code for the original manuscript, I mistakenly referred to results from very early days of the project, from two years ago, which is further illustrated in Figure 1 and Figure 3 below for additional context. As you can see in Figure 1 and Figure 3, the files used are directly taken from Jan 2022. Honestly, there have been so many bug fixes and code improvements since that very old "data" was generated that there is no telling what the numbers in that file actually mean. Furthermore, you could see the plotting code itself has not been modified since May 2023 when I commenced debugging for the first revision. As acknowledged above, during that time I mistakenly thought this was a Pandas bug because other than that issue which is reflected on my Github code change, I did not find anything else in my link prediction code. I never thought the problem could be in my plotting code.

Through the provided screenshots from my local computer and all corresponding plotting code,

I want to show to you that the Figure 6, which is called “side by side auc.pdf” on my local end, has remained unchanged since January 2023. This evidence is also available in the uploaded folder named “Mistake and Proof” in the GitHub repository, which the reviewer could feel free to check the time stamp on themselves if they so desire.

Indeed, and in the interests of fullest possible disclosure, I note something the reviewer might not have previously noticed, but I apparently made another mistake in that plotting code as well by utilizing the same wrong and old result to plot the results that appear in both the completely unobserved and partially observed cases in the original submission, as evidenced in Figure 2. That is, upon closer examination, it becomes obvious that the AUC plot exhibits identical shapes for both settings in the original submission.

The new revision does not make any further alterations to Figure 6, as I have thoroughly verified the correctness of both the link prediction code and the plotting code. Once again, all example codes are accessible online on my GitHub repository. Moreover, I have further updated the GitHub repository prior to the present revision to make it easier for users to download and perform the full Ensemble-Sequential-Stacking method, which requires running the other predictors that are parts of others’ code, so I had not included it previously; but do so now in the interests of making a more usable code of the full ensemble method. I encourage the reviewer to examine the small networks if any questions or doubts persist.

R1.4

Consequently, despite my utmost respect for the senior authors' reputation, I must maintain my stance of not endorsing the publication of this manuscript in Nature Communications. Despite my professional admiration, the rigor of the research appears insufficient to warrant acceptance at this juncture.

We acknowledge the reviewer’s concerns and deeply value their profound expertise and rigorous approach to this review. We are committed to demonstrating that we have presented ample supporting evidence to address the reviewer’s reservations.

I genuinely appreciate the reviewer for bringing this issue to our attention. Without their diligent scrutiny, I might not have detected the error in the plotting code. I sincerely regret any inconvenience this may have caused and am immensely grateful for the reviewer’s unwavering dedication. I wholeheartedly value this constructive feedback and am dedicated to incorporating this invaluable lesson into all my future endeavors.

```
In [2]: import numpy as np
import matplotlib.pyplot as plt

data_list1 = ["bionet1", "bionet2", "chess", "bitcoin", "collegemsg", "obitcoin", "obrazil", "radoslaw", "london", "mit"]
data_list2 = ["ant1", "ant2", "ant3", "ant4", "ant5", "ant6"]
data_list3 = ["fbforum", "fbmsg", "emaildnc"]

data_list = data_list1+data_list2+data_list3

r1 = np.loadtxt("real_auc1_20.txt")
r2 = np.loadtxt("real_auc2_20.txt")
r3 = np.loadtxt("real_auc3_20.txt")
r4 = np.loadtxt("real_lstm.txt")
r5 = np.loadtxt("real_auc5_20.txt")
r6 = np.loadtxt("real_auc6_20.txt")

new_data_list1 = ["chess", "obrazil", "bionet1", "bitcoin", "emaildnc", "bionet2",
                 "obitcoin", "london", "collegemsg", "fbmsg", "radoslaw", "fbforum", "mit",
                 "ant1", "ant2", "ant3", "ant4", "ant5", "ant6"]

nr1 = []
nr2 = []
nr3 = []
nr4 = []
nr5 = []
nr6 = []
for item in new_data_list1:
    idx = data_list.index(item)
    nr1.append(r1[idx])
    nr2.append(r2[idx])
    nr3.append(r3[idx])
    nr4.append(r4[idx])
    nr5.append(r5[idx])
    nr6.append(r6[idx])

In [3]: path = r"C:\Users\hexie\OneDrive\Documents\Temporal Link Prediction\setting2\"
sr1 = []
sr2 = []
sr3 = np.loadtxt(path + "real_lstm.txt")
for name in data_list:
    q = np.loadtxt(path + str(name) + "_0_auc.txt")
    q0 = q[0]
    q1 = q[1]
    sr1.append(q0)
    sr2.append(q1)

snr1 = []
```

Figure 1: Screenshot of the code loading the same files, you can see the last time it was revised was 5/10/2023, as I am trying to find the bug but failed, because the variables r1, r2, r3, r4, r5, r6 are still not changed to new files, which I overlooked.

Figure 2: Screenshot of the code loading the same files, but this time note that I am using nr1 and nr2 and nr3 to plot the completely unobserved as well as the partially observed case even though it should sr1, sr2, sr3 which I have loaded from the previous setting 2 document. This is the reason why the original version's plot is entirely wrong because the plotting code is loading the wrong file.

Name	Status	Date modified	Type	Size
side_by_side auc		5/10/2023 1:56 PM	IPYNB File	122 KB
AUC-COUNTER-FOURAUUC-FINAL-WITH-...		5/7/2023 1:44 PM	IPYNB File	253 KB
6 auc plot setting2		5/7/2023 1:44 PM	IPYNB File	47 KB
AUC-COUNTER-FOURAUUC		5/7/2023 1:13 PM	IPYNB File	225 KB
fig-fake1-fancy		1/23/2023 8:16 PM	IPYNB File	126 KB
fig-fake2-fancy		1/23/2023 8:16 PM	IPYNB File	107 KB
side_by_side auc		1/23/2023 7:59 PM	Adobe Acrobat D...	20 KB
print_precision_recall_fake_setting2		11/3/2022 12:24 PM	IPYNB File	8 KB
print_precision_recall_real_setting2		9/13/2022 6:06 PM	IPYNB File	9 KB
print_precision_recall_real		9/13/2022 6:01 PM	IPYNB File	19 KB
print_precision_recall_fake1		9/13/2022 6:01 PM	IPYNB File	11 KB
print_precision_recall_fake		7/20/2022 4:47 PM	IPYNB File	8 KB
produce_auc_values		7/18/2022 6:26 PM	IPYNB File	137 KB
two_sample_ttest		3/14/2022 5:26 PM	IPYNB File	15 KB
pvalue		2/9/2022 9:37 PM	IPYNB File	79 KB
LTP_science_template_CCP		2/8/2022 5:41 PM	Adobe Acrobat D...	467 KB
LTP_science_template		2/8/2022 5:07 PM	Adobe Acrobat D...	428 KB
pvalue		2/3/2022 8:28 PM	Adobe Acrobat D...	2 KB
real_lstm		1/31/2022 7:17 PM	Text Document	1 KB
real_auc1_20		1/31/2022 7:16 PM	Text Document	1 KB
real_auc2_20		1/31/2022 7:16 PM	Text Document	1 KB
real_auc3_20		1/31/2022 7:16 PM	Text Document	1 KB
real_auc4_20		1/31/2022 7:16 PM	Text Document	1 KB
real_auc5_20		1/31/2022 7:16 PM	Text Document	1 KB
real_auc6_20		1/31/2022 7:16 PM	Text Document	1 KB
STAT		1/21/2022 11:55 AM	NPY File	2 KB
TEMP		1/21/2022 11:55 AM	NPY File	2 KB
figures		5/7/2023 1:20 PM	File folder	

Figure 3: Screenshot of the time stamp on the files. As you can see the file *sidebysideauc.ipynb* was only modified on May 10, 2023 because I was trying to find the bug, but the loaded files *realauc120.txt*, etc. are not changed since January 31, 2022. These files are also attached. Note also that *sidebysideauc.pdf* is not changed after January 23, 2023, which is also attached as a supplement, as you can see, it is exactly the same as the one in the original version of the submission.

Reviewers' Comments:

Reviewer #1:

Remarks to the Author:

The authors have made a sincere effort to identify the root cause of the inconsistent results observed in their initial and subsequent paper submissions. I appreciate their diligence in this regard.

It is now sufficiently clear that the inaccuracies in the original submission's stacking results stemmed from the utilization of older or preliminary runs of the method. Based on the evidence presented in their rebuttal, I am reasonably convinced that this might have been the source of the stacking results' error.

Regrettably, this explanation regarding the stacking results still leaves some questions unanswered. It doesn't clarify why benchmark algorithms, in certain cases, transitioned from nearly flawless performance in the first submission to results akin to random guessing in later submissions. I previously noted this phenomenon concerning the T-SBM method in the context of chess and obrazil networks. In both cases, the initial submission demonstrated almost perfect performance by T-SBM, while subsequent submissions exhibited poor performance, even comparable to random guessing.

I attempted to examine the authors' code in an effort to reproduce the reported results for the benchmark algorithms, particularly T-SBM. Unfortunately, I found no clear indication of these experiments in their GitHub repository. The only SBM code I found was within the "Ensemble_final_edition," and I encountered difficulties when attempting to download and preprocess the data for rerunning the code to evaluate each algorithm's performance separately. Additionally, upon reviewing the code, further questions arose. It appears that the T-SBM code necessitates information about all existing and non-existing links, referred to as the 'list of true edges' and 'list of false edges' in the code. For a network such as chess, with over 7,000 nodes and nearly 100 layers, this would entail providing the algorithm with over 2 billion edge values. How is this accomplished precisely? Can the algorithm effectively process such massive amounts of data?

Given these lingering uncertainties, I am not yet prepared to endorse publication in Nature Communications. However, I want to reiterate that my evaluation is primarily based on the inconsistencies observed during the review process and does not reflect any judgment on the scientific quality of the researchers involved. We all make mistakes and I feel that our role, when acting as reviewers, is to try to avoid that those mistakes go unnoticed.

Review 1

R1.1

The authors have made a sincere effort to identify the root cause of the inconsistent results observed in their initial and subsequent paper submissions. I appreciate their diligence in this regard.

It is now sufficiently clear that the inaccuracies in the original submission's stacking results stemmed from the utilization of older or preliminary runs of the method. Based on the evidence presented in their rebuttal, I am reasonably convinced that this might have been the source of the stacking results' error.

We appreciate your recognition of our efforts to identify the inconsistencies in our submissions. Your careful reading and feedback have been instrumental in refining our work, and we are committed to addressing these issues.

R1.2

Regrettably, this explanation regarding the stacking results still leaves some questions unanswered. It doesn't clarify why benchmark algorithms, in certain cases, transitioned from nearly flawless performance in the first submission to results akin to random guessing in later submissions. I previously noted this phenomenon concerning the T-SBM method in the context of chess and obrazil networks. In both cases, the initial submission demonstrated almost perfect performance by T-SBM, while subsequent submissions exhibited poor performance, even comparable to random guessing.

Regrettably, since the original Figure 6 resulted from a plotting code error, loading outdated results files that were so preliminary that we have no way of knowing what was actually being saved. Unfortunately, because of the extended time frame since those outdated results were generated, we are unable to access those early versions of our then-in-development codes, hindering our ability to pinpoint the sources of specific differences further. That said, we understand the reviewer's natural curiosity here, and in looking closely at the results, we believe it is vital to emphasize that the T-SBM results have not entirely decreased. Indeed, on the positive side, the correction of the plotting bug has resulted in a significant increase in T-SBM performance in other networks, such as the 6 ants networks, where values rose from ~ 0.6 to ~ 0.8 . Also, notably, for the obrazil network, AUC scores became lower for all methods, indicating the wide relevance of the errors we made in that first plot. We sincerely regret this oversight and rectified the plotting code error in preparing the correct results. We have repeatedly checked and re-checked that we are now presenting the correct results, and that they are reproducible; we encourage you to refer to the correct Figure 6 for accurate results.

R1.3

I attempted to examine the authors' code in an effort to reproduce the reported results for the benchmark algorithms, particularly T-SBM. Unfortunately, I found no clear indication of these experiments in their GitHub repository. The only SBM code I found was within the "Ensemble

final edition,” and I encountered difficulties when attempting to download and preprocess the data for rerunning the code to evaluate each algorithm’s performance separately.

Given the previous comment, we understand the reviewer’s particular and understandable interest in being able to reproduce the reported results for the benchmark algorithms. We do want to stress, however, that we are now in the realm of dealing with difficulties about running other peoples’ codes, not ours. That said, since the primary message of our paper includes the ensemble learning results using these other existing benchmarks, we believe it is important for users (not just the reviewers) to be able to reproduce all of our results, including those we obtained from others’ benchmarks.

So motivated by the reviewer’s feedback about result reproduction and agreeing that it is of essential importance to demonstrate as such, we have diligently and significantly revised the associated GitHub repository with a complete set of instructions about how to reproduce all or, in some cases, only some selected parts of our results. In particular, we are unable to generate results with the E-LSTM-D benchmark outside the narrow range of Python versions 3.6.7–3.7, because of its dependencies on other packages over which we have no control because this isn’t our code. Our full set of instructions thus includes multiple options about which methods are and are not used under different settings. Importantly, we recruited a set of volunteers from outside our respective labs who diligently tested our instructions for running the codes on Mac, Linux, Windows, and in Colab. Each of these volunteers has reported successful builds and experimental runs on all platforms (up to limitations with the E-LSTM-D benchmark, as described). To facilitate result reproduction, please refer to the updated GitHub instructions and README. Please ensure the installation of environment files before proceeding. Our experiment setup for E-LSTM-D and T-SBM can be found in the Methods section, and we have also included that information in our GitHub if using the default setting. As a note for easier access and your convenience, we have also attached our GitHub README as a supplement to this response.

Additionally, we note that T-SBM has been updated since the version we used from their GitHub repository. We also note that there is a typo in that older version from their GitHub, where they mislabeled the channel and target node, but we fixed it in our code before using it in our experiments and including it in our files.

R1.4 *Additionally, upon reviewing the code, further questions arose. It appears that the T-SBM code necessitates information about all existing and non-existing links, referred to as the ‘list of true edges’ and ‘list of false edges’ in the code. For a network such as chess, with over 7,000 nodes and nearly 100 layers, this would entail providing the algorithm with over 2 billion edge values. How is this accomplished precisely? Can the algorithm effectively process such massive amounts of data?*

We understand the reviewer’s concern here, and this is precisely why, in our approach, we have introduced flow variable q and search variable u . In particular, u determines how far back we want to go back in time to make the prediction on a given layer. As noted in the paper, we have defaulted all of our experiments to have $u = 6$ and $q = 3$, which, in turn, have made the network considerably smaller. Recognizing the importance of this point, we have more strongly emphasized it in a small number of new edits to the Introduction, and have additionally highlighted this relevant information in our paper regarding u and q repeatedly. We have also discussed the choice of these two parameters in the Discussion section, as we expect it could be useful to make different choices for different real-world networks, and could potentially be an interesting future direction of study. But for the scope of this paper, due to

the high computational time already undertaken, including that of T-SBM and thus the full Ensemble-Sequential method, we have set them to be default values for all of our experiments.

Additionally, we note the importance of the comment here about the storage on the ‘list of true edges’ and ‘list of false edges.’ For the larger networks, we resolve this through sampling from the original network, and in our edits we have made an additional note about this in the Introduction as well. For convenience, we have also highlighted this section of the Methods for easier reference, though we emphasize that we have not edited this section from the previously submitted version.

That all said, though we have not focused on the computational time differences in the paper, we note briefly here that for a small network with 200 nodes, ‘T-SBM’ usually takes around 2 hours to complete the calculation. In contrast, both ‘E-LSTM-D’ and ‘Top-Sequential’ took around 5 minutes. For larger networks like ‘chess,’ even when only considering the first 7 layers (because we set $u = 6$), we chose to let T-SBM run on a cluster overnight, and thus, to run it locally could be quite a challenging task indeed.

R1.5 *Given these lingering uncertainties, I am not yet prepared to endorse publication in Nature Communications. However, I want to reiterate that my evaluation is primarily based on the inconsistencies observed during the review process and does not reflect any judgment on the scientific quality of the researchers involved. We all make mistakes and I feel that our role, when acting as reviewers, is to try to avoid that those mistakes go unnoticed.*

We sincerely value the reviewer’s caution and assistance. We thank you again. We hope that our response here, coupled with the updated instructions on GitHub, reflects our commitment and diligence to get this right, and we hope these additional answers and efforts can convince the reviewer to endorse the paper.

Sequential Stacking Link Prediction ↗

Sequential Stacking Link Prediction Algorithms for Temporal Networks ↗

This is the GitHub repo accompany the paper by Xie He, Amir Ghasemian, Eun Lee, Aaron Clauset, Peter Mucha. The paper is currently under revision at Nature Communications.

Please cite the paper when using the data or code. See License Information for more details on Usage.

To ensure reproducibility, the below information has been tested and successfully run by volunteers who read the GitHub and then experimented on Linux, Mac, Windows, and Google Colab(except the E-LSTM-D, because it does not work for Google Colab, but should work on all other platforms with the constraints described below).

System Requirements ↗

To reproduce all results from our experiments, you will need at least Python 3.6.7 (but E-LSTM-D will not work above 3.7) and a few packages installed (see the environment file for specific details).

You can check your python version with

```
$ python --version
```

Alternatively, if you wish to run only the Top-Sequential or T-SBM method with the topological features, you could instead do:

```
pip install Imbalanced-learn scipy numpy pandas networkx scikit-learn
```

If you further wish to run Time Series, then you should also install:

```
pip install statsmodels
```

If you run into trouble with the original E-LSTM-D GitHub, but you want to run the E-LSTM-D and the full Ensemble-Sequential method, then you should first make sure you have python 3.6.7+ but no more than 3.7. Then you could try do the following to create the required environment for the code (**Only if you want to use the full Ensemble-Sequential**, because of the dependency of E-LSTM-D, Time Series, and T-SBM).

```
pip install tensorflow==1.14.0 keras==2.2.4 Imbalanced-learn==0.8.1 scipy==1.5.4 scikit-learn==
```

The above environment has been tested to build successfully and run all the following experiments successfully on all the popular platforms and should work for Windows, Mac OS, and Linux, if installed correctly. (numpy 1.19.5 is also okay)

Data Input Format Requirements ↗

Throughout the usage of this repo, please make sure your nodes are labeled from 0 to N-1 with integer as their index. This applies to real-world networks too. Node string names are not yet supported.

The input files of the network should be put in a folder that is the name of that network. Each temporal layer should be separated into different files starting at number 1.

In other words, in the directory `fake110`, it should contain `fake110_1.txt`, `fake110_2.txt`, etc.

The content of the txt file should be: `source_node_idx target_node_idx` for each edge on each line. See the example synthetic dataset as a reference.

For simplicity, here I only describe the process for the partially observed case, the completely unobserved case is done in the exact same setting, but with slightly different named python files (usually there's the word complete in the file name).

To run only the Top-Sequential Experiments

The best way to run only the Top-Sequential Experiment is to follow the `example.py` file. In order to run it, first you have to unzip the `community_label_TSBM.zip`.

```
$ python example.py
```

Change the variables and/or numbers in `example.py` to change the corresponding variables in the paper.

Note that you have to manually determine the number of layers you want the algorithm to work with.

- The search variable `u` could be found and replaced in `edges_orig = edges_orig[0:u]` (6 in all of our experiments)
- The flow variable `q` could be found and replaced in `predict_num = q` (3 in all of our experiments)

Running `example.py` (which contain two functions) will generate two AUC scores, accordingly with the partially observed case and the completely unobserved case in the paper.

To run Top-Sequential and the T-SBM without the whole Ensemble-Sequential method (NOT VERY RECOMMENDED, but doable)

There's currently no way to run T-SBM individually in this directory, because arguably the best way to run it individually will be to run it through its original GitHub T-SBM.

Running the experiments will take a while depending on your hardware. In particular, T-SBM could be a bit slow even for smaller networks.

If you do not wish the run the full E-LSTM-D and Time Series, but are only interested in the topological feature and T-SBM, you could simply navigate to the folder, and run:

```
$ cd ensemble_with_others/Ensemble_final_edition
$ python data_runner.py # this will create the T-SBM features (which would be an edge
indicator) and the Topological features
```

This will give you all the feature matrix you need to further use your preferred algorithm.

If done correctly, you should see: `"for_sbm"`, `"feature_metrics"`, `"results"`, `"edge_tf_true"`, `"edge_tf_tr"`, `"ef_gen_ho"`, `"ef_gen_tr"`.

You need to change the variable `feat_path` in the file `calculate_different_AUC.py` to your own feature path before proceeding if you have **NOT** run the other two commands (see below for details).

Because now that the order is disturbed, you need to load the feature matrix from the folder named `feat_path = "./ef_gen_tr/"` for the training matrix, and then `feat_path = "./ef_gen_ho/"` for the hold out matrix. Note also that you need to rename them in order to make `calculate_different_AUC.py` recognize them. To be even more specific, the input of the `calculate_different_AUC.py` requires four different things: `df_t_tr` for the true training edges, `'df_f_tr'` for the false training edges, and `df_t_ho` for the true hold out edges, and `df_f_ho` for the false holdout edges. Thus you should make sure that these files all exist before you do anything else. Likely they live in the previously mentioned folders, probably named to just be "df_t" and "df_f".

To help the reader make this process easier, I have built the function `TOP_TSBM_postprocess.py` as a help function to make this process easier. Please put in the same folder as `data_runner.py`. Please be careful with this function as it **WILL OVERWRITE** the final output if you directly run it after you have successfully built everything. This file should be run **BEFORE** you call `python calculate_different_AUC.py`.

```
$ python TOP_TSBM_postprocess.py
```

Thus, it is highly recommended that you finish the whole process first, or **at least finish the Time-Series part first** before you proceed to call `calculate_different_AUC.py`.

Once you are sure that you have the feature matrix you want in the folder you want it in, go ahead and call:

```
$ python calculate_different_AUC.py
```

If you have run the other two in the order described above, then you can ignore the whole section above and directly call it, as this function is meant to be called after all the feature matrices have been generated.

Very Importantly, in the file `calculate_different_AUC.py`, the main loop contain a variable named `choice`.

The choice `0` gives you Top-Sequential AUC, choice `1` gives you Time-Series, choice `2` gives you T-SBM, choices `3` gives you E-LSTM-D, and choice `4` gives you Ensemble-Sequential-Stacking, just like what is described above.

And in the case you have **NOT** run neither Time-Series nor E-LSTM-D, you only have the choice of `0` and `2`. Any other option will likely give you an error message.

To run Top-Sequential, T-SBM and Time-Series without the whole Ensemble-Sequential method (RECOMMENDED Only if having a lot of trouble with E-LSTM-D) 🔗

```
$ cd ensemble_with_others/Ensemble_final_edition
$ python data_runner.py # this will create the T-SBM features (which would be an edge
```

indicator) and the Topological features

```
$ python process_ts.py # this will create the time series features and add them to the end of the previous features.
```

If done correctly, there will be a folder named "all_features" appearing after the run. Replace the variable `feat_path` value in the file `calculate_different_AUC.py` to `all_features` and run

```
$ python calculate_different_AUC.py
```

All the rest will be exactly the same as described in the above section.

To run the full Ensemble-Sequential Experiment (HIGHLY RECOMMENDED if E-LSTM-D works out fine) ↗

Running the experiments will take a while depending on your hardware. In particular, T-SBM could be a bit slow even for smaller networks.

To run the individual benchmarking methods:

1. Download and install the code and relevant packages from: E-LSTM-D
2. Download and install the code and relevant packages from: T-SBM
3. Make sure you have installed the required environment and packages.

To run the full Ensemble-Sequential experiment, first, you have to run the modified E-LSTM-D codes provided in this repository in order to get the features and AUC scores from it. See System Requirements for how to install the environments correctly.

```
$ cd ensemble_with_others/E-LSTM-D/Partially-observed
$ python convert_partial.py
$ python calculate_elstmd.py
$ python generate_output.py
```

This will in turn gives you a full feature matrix from E-LSTM-D and a folder named "lstm_feat", which you could use to stack with the topological features extracted with Top-Sequential method.

If you wish to get the AUC scores for E-LSTM-D only, stop here.

Then, please go ahead and copy and paste the folder `lstm_feat` to be under the same directory that you are planning to conduct your full Ensemble-Sequential method. (In the code, I intentionally avoid directly putting it under that folder to avoid confusion about where that output folder comes from.)

After that, navigate to the folder `ensemble_with_others/Ensemble_final_edition/`, which is also the default folder that you should be pasting to.

Once inside the folder you have to first generate the feature matrix for the dataset first. You can do this by:

```
$ python data_runner.py # this will create the T-SBM features (which would be an edge index)
$ python process_ts.py # this will create the time series features and add them to the end of t
$ python create_lstm_df.py # this will create the LSTM features. Omit this step if you have not
```

If done correctly, you should be seeing folders named "finalized_all_features", "all_features", "lstm_feat", "for_sbm", "feature_metrics", "results", "edge_tf_true", "edge_tf_tr", "ef_gen_ho", "ef_gen_tr".

Then you could go ahead and call:

```
$ python calculate_different_AUC.py
```

This will give you the complete AUC scores result of the dataset you desired. If left not touched, it will output to the folder named `full_results_final`.

Very importantly, the AUC scores order that you will end up getting after the partially observed case should be in the following order:

```
auc_methods = ['Top-Sequential-Stacking', 'Time-Series', 'Tensorial-SBM', 'E-LSTM-D',  
'Ensemble-Sequential-Stacking',]
```

The AUC scores order that you will get after the completely unobserved case will be the same order, except that you will ignore the third column, `Tensorial-SBM`, because that would be a meaningless result that is repeating the partially observed case.

Note also: feel free to use this ensemble learning method stacked with other features of your liking. Theoretically, any features that could be generated with a partially observed network would work with that case, and note also that the completely unobserved case would require features that could be generated from the previous time layers.

If there are any questions, feel free to leave a message on GitHub or email directly.

To run the benchmarking methods mentioned in the paper individually ↗

For E-LSTM-D:

1. Download and install the code and relevant packages from: E-LSTM-D
2. Either you could then run their code directly to calculate the AUC.
3. Or you could directly run the full Ensemble-Sequential code, which automatically generates the AUC scores after the full-run.

For Tensorial-SBM:

1. Download and install the code and relevant packages from: T-SBM

2. Either you could then run their code directly to calculate the AUC.
3. Or you could directly run the full Ensemble-Sequential code, which automatically generates the AUC scores after the full-run.

For Time Series:

See above for detailed description.

```
$ cd ensemble_with_others/Ensemble_final_edition
$ python data_runner.py # this will create the T-SBM features (which would be an edge
indicator) and the Topological features
$ python process_ts.py # this will create the time series features and add them to the end
of the previous features.
```

Synthetic Datasets ↗

The example runs could be found in `example.py`, which runs through one of the 90 synthetic network datasets we created. To run through the synthetic networks, please download them through the Google Drive Link here. Once downloaded, go ahead and extract the folder into the same folder under `TOLP.py` and change the path name in the `example.py` and/or modify to your liking.

Note that the naming of the synthetic networks could be very confusing. Here we list the naming pattern for both types of synthetic network so that the readers are not confused. We did the naming this way to avoid long and arduous names of the files. For the naming convention, see the functions in the python file `translate.py` for specific details.

Real World Datasets ↗

The real world networks could be found under the following links. Due to copyright reasons, we will only show the link to download them. The following is taken from ICON:

<https://icon.colorado.edu/#!/networks>

- chess: Search for Kaggle chess players (2010) on : <https://icon.colorado.edu/#!/networks>
- bitcoin: Bitcoin Alpha trust network (2017): <https://snap.stanford.edu/data/soc-sign-bitcoinalpha.html>
- obitcoin: Bitcoin OTC trust network (2017): <https://snap.stanford.edu/data/soc-sign-bitcoinotc.html>
- obrazil: Brazilian prostitution network (2010): <http://konect.cc/networks/escorts/>
- london: London bike sharing (2014): <https://github.com/konstantinklemmer/bikecommclust>
- mit: Search for Reality mining proximity network (2004) on: <https://icon.colorado.edu/#!/networks>
- radoslaw: Search for Manufacturing company email (2010) on: <https://icon.colorado.edu/#!/networks>

The following is taken from network repository:

- ant1-ant6: <https://networkrepository.com/asn.php> (see insect-ant-colony)

- emaildnc: <https://networkrepository.com/email-dnc.php>
- fbforum: <https://networkrepository.com/fb-forum.php>
- fbmsg: <https://networkrepository.com/fb-messages.php>

The following is given to us by the authors, special thanks to the authors for sharing the data.

- bionet1-2: https://www3.nd.edu/~tmilenko/software_data.html
- Khaliq Newaz and Tijana Milenkovic (2020), Improving inference of the dynamic biological network underlying aging via network propagation, IEEE/ACM Transactions on Computational Biology and Bioinformatics, DOI: 10.1109/TCBB.2020.3022767.

Acknowledgements

We thank Marya Bazzi, Lucas Jeub, Roxana Pamfil, and Mason A. Porter for helpful discussions and conversation; and Khaliq Newaz and Tijana Milenkovic for providing the bionet datasets. We are grateful for the use of the high performance computing clusters at the University of North Carolina at Chapel Hill (longleaf) and Dartmouth College (discovery7). A special thanks to Junyi Cheng for helping with the graphical design of Figure 1. Special thanks to Jonathan T. Lindbloom, Lizuo Liu, and Ryan Maguire for their help during the progress of this project. This work is supported in part by the Army Research Office under MURI award W911NF-18-1-0244 (X.H. and P.J.M.), the National Science Foundation under Grant No.~2030859 to the Computing Research Association for the CIFellows Project (A.G.), and the National Research Foundation of Korea (NRF) grant funded by the Korea government(MSIT) (No.~RS-2022-00165916) (E.L.). The content is solely the responsibility of the authors and does not necessarily represent the official views of any agency supporting this research.

Previous Mistakes

You could also find the past bugged version of the code both in the same folder and on GitHub for debugging purposes. The noticeable change could be found in the GitHub history. There might still be unfound BUGs, email me or leave a message as you see fit.

Releases

No releases published

Create a new release

Packages

No packages published

Publish your first package

Languages

Reviewers' Comments:

Reviewer #1:

Remarks to the Author:

I thank the authors once more for their efforts to clarify the inconsistencies between different versions of their manuscript.

After this new revision and rebuttal, doubts remain. In particular, the authors "are unable to access those early versions of our then-in-development codes, hindering [their] ability to pinpoint the sources of specific differences further." (Note that those might have been "in-development" codes, but they were the ones used to create the plots in the original submission!) Therefore, we are definitively left with no explanation as to why some algorithms' performance *decreased* dramatically after fixing whatever bugs there might have been in the original code. The authors argue that the performance of those algorithms also increased in some instances, but this is entirely irrelevant, since improving performance is exactly what one expects when fixing a bug. As I mentioned in a previous report, I still do not understand what kind of "bug" can make an algorithm, which now seems to work very poorly, work almost perfectly.

My only speculation in this respect is that the authors might have changed some of the experimental conditions at some point (perhaps the values of q , u , or the sampling for building the train sets), which would raise questions about the robustness of the results.

In any case, the authors have now made a good job at making their results reproducible by providing thorough documentation in their code repository. Now, interested readers from the community will at least be able to exactly pinpoint what was done, and perhaps amend or build on their work and conclusions.